

# Uncertainty assessment and applicability of an inversion method for volcanic ash forecasting

Birthe Marie Steensen[1], Arve Kylling[2], Nina Iren Kristiansen[2], Michael Schulz[1]

[1]Research department, Norwegian Meteorological Institute, Oslo, 0131, Norway
[2]Atmosphere and Climate Department, Norwegian Institute for Air Research (NILU), Kjeller, 2007, Norway

*Correspondence to*: Birthe Marie Steensen (birthe.steensen@met.no)

**Abstract.** Significant improvements in the way we can observe and model volcanic ash clouds have been obtained since the 2010 Eyjafjallajökull eruption. One major development has been data assimilation techniques, which aim to bring models in closer agreement to satellite observations and reducing the uncertainties for the ash emission estimate. Still, questions
remains to which degree the forecasting capabilities are improved by inclusion of such techniques are and how these improvements depend on the data input. This study exploits how different satellite data and different uncertainty assumptions of the satellite and a priori emissions affect the calculated volcanic ash emission estimate, which is computed by an inversion method that couples the satellite and a priori emissions with dispersion model data. Two major ash episodes over four days in April and May of the 2010 Eyjafjallajökull eruption are studied. Specifically, inversion calculations are
done for four different satellite data sets with different size distribution assumptions in the retrieval. A reference satellite data set is chosen and the range between the minimum and maximum 4 day average load of hourly retrieved ash is 121 % in April and 148 % in May, compared to the reference. The corresponding a posteriori maximum and minimum emission sum found for these four satellite retrievals range from 26 % and 47 % of the a posteriori reference estimate for the same two periods. Varying the assumptions made in the satellite retrieval therefore translates into uncertainties in the calculated emissions and
the modelled ash column loads. By further exploring the weighting of uncertainties connected to a priori emissions and the other-than-size uncertainties in the satellite data, the uncertainty in the a priori estimate is found to have an order of magnitude more impact on the a posteriori solution compared to the other-than-size uncertainties in the satellite. Part of this is explained by a too high a priori estimate used in this study that is reduced by around half in the a posteriori reference estimate. Setting large uncertainties connected to both a priori and satellite input data is shown to compensate each other.
Because of this an inversion based emission estimate in a forecasting setting needs well tested and considered assumptions on uncertainties for the a priori emission and satellite data. The quality of using the inversion in a forecasting environment is tested by adding gradually, with time, more observations to improve the estimated height versus time evolution of Eyjafjallajökull ash emissions. We show that the initially too high a priori emissions are reduced effectively when using just 12 hours of satellite observations. More satellite observations (>12h), in the Eyjafjallajökull case, place the volcanic
injection at higher altitudes. Adding additional satellite observations (>36h) changes the a posteriori emissions to only a small extent for May and minimal for the April period, because the ash is dispersed and transported effectively out of the





domain after 1-2 days. A best-guess emission estimate for the forecasting period was constructed by averaging the last 12 hours of the a posteriori emission. Using this emission for a forecast simulation performs better especially compared to model simulations with no further emissions over the forecast period in the case of a continued volcanic eruption activity. Because of undetected ash in the satellite retrieval and diffusion in the model, the forecast simulations generally contain

more ash than the observed fields and the model ash is more spread out. Overall, using the a posteriori emissions in our model reduces the uncertainties connected to both the satellite observations and the a priori estimate to perform a more confident forecast in both amount of ash released and emission heights.

## 1 Introduction

The fine ash fraction (ash particles with diameter < 64 µm) of tephra from volcanic eruptions can be transported over large

distances and cause jet engine malfunction and damages to windshields (Casadevall, 1994). Both the 2010 April and May Eyjafjallajökull eruption and the May 2011 Grimsvötn eruption caused flight delays and cancellations leading to economical loss (European Commission, 2011). Although satellite observations can show snapshots of the current horizontal extension of ash, volcanic ash transport and dispersion models (VATDMs) are needed to forecast the dispersion of the volcanic clouds. The source term needed to run these forecasts can be highly uncertain. Stohl et al. (2011) presents an inversion method to

calculate a source term constrained by satellite observations, using a priori emission estimates and model simulations. This inversion technique has been successfully applied to calculate ash emissions from the Eyjafjallajökull and Grimsvötn eruptions as well the 2014 Kelut eruption (Stohl et al., 2011; Kristiansen et al., 2012; Moxnes et al., 2014; Kristiansen et al., 2015). The method has also been applied to volcanic eruptions with $SO_2$ emissions (Kristiansen et al. 2010; Eckhardt et al. 2008).

The satellite data, a priori and model input data required by the inversion algorithm all have assumed uncertainties connected to them that weight the inversion calculations. Both the assumed a priori and satellite uncertainties used in the studies mentioned above varies from around 100% of the input data values and downwards to zero or a minimum value based on the confidence of the a priori emission and satellite data available for the three eruption cases. For the Kelut eruption however, where the eruption reached the stratosphere, the a priori emission estimate was highly unreliable so the uncertainty was set to

1000% of the assumed a priori emission values to make the result be almost exclusively driven by the satellite data. Eckhardt et al (2008) found that using a zero and constant a priori estimate gave similar a posteriori estimates for the Jebel at Tair 2007 eruption. This highlights that uncertainty settings in the inversion are case dependent. In this study, inversion calculations with different assumed uncertainties are presented to increase understanding of the effects on the a posteriori emissions.

Boichu et al. (2013) investigated the $SO_2$ emissions of the 2010 Eyjafjallajökull eruption in early May by a similar inversion method and found that $SO_2$ emission estimate calculations by only a single satellite image gave consistent results for young plumes, but showed increased uncertainty as the plume evolved over time. A more accurate emission term are found by



assimilating more satellite observations further into the period and therefore reducing the forecasting time. Wilkins et al. (2016a) used an insertion method for ash forecasting by initializing a dispersion model with ash layers derived from SEVIRI (Meteosat Second Generation Spinning Enhanced Visible and Infrared Imager) satellite retrievals. The study found that the model field calculated by including up to six satellite observations gave a broader and more extensive ash cloud, which

compared worse to the 8 May 9 UTC satellite observation, than a single satellite retrieval inserted six hours before the observation time. The ash cloud found by several retrievals is considered a more conservative choice for giving commercial air traffic advice however as it includes ash that may not be captured by a single observation.

In the previous studies using the inversion method by Stohl et al. (2011), the a posteriori emission estimates were calculated after the eruption had ceased and using all satellite data available for the entire eruption period. However, in this study more

satellite observations will be added gradually to the inversion algorithm to simulate a real forecast scenario. The purpose of using the inversion method is to make the model simulated ash with the inversion derived a posteriori source emission term more alike the observed ash column loads, as compared to model simulations with source terms calculated by empirical plume height relationships like the one given in Mastin et al. (2009) (used here as a priori emission). The inversion algorithm only calculates a constrained source term up until the start of the forecast, as it requires satellite observations. For emissions

during the forecast period one has to use other methods 1) assume no further emissions 2) use the latest a priori emission from Mastin et al. (2009), or 3) average of the last hours of the a posteriori from the inversion. As the latter option includes some information from the satellite observations that limits the uncertainty, a 12 hours average will be tested here as opposed to zero forecast emission.

Meteorological clouds that contain ice, super-cooled droplets or unfrozen cloud droplets decrease the ability to identify ash

in satellite retrievals, or retrieve higher concentrations than what is the truth (Prata and Prata, 2012, Kylling et al., 2015). Retrieval of ash from one single satellite image of the cloud is therefore more uncertain than a series of retrievals covering a longer time period. Hourly SEVIRI satellite retrievals are used in this study, and weaknesses in the satellite retrievals are explored further by differentiating pixels where no ash is detected and unclassified pixels where it is uncertain if the pixels contain ash.

The aim of this study is to use the inversion method by Stohl et al. (2011) in a forecasting setting and investigate how changes in input influence emission estimate results. Two four days periods in April and May of the 2010 Eyjafjallajökull eruption are studied. During the first period from 14 to 18 April, an ash cloud is transported over Central Europe originating from ash emitted on 14 and 15 April, while a smaller amount is released on 17 April. The second period studied covers 5 to 9 May when more ash was emitted again after a period with low emissions. The ash was transported south and entrained in a

high pressure system causing the ash cloud to persist over the North Atlantic and stay in the domain over the whole period. More satellite observations are therefore available for this episode.

The paper is structured as follows: section two gives a short description of the inversion method, the model and satellite data used in this study, as well as the structure, amplitude and location (SAL) scoring method (Wernli et al., 2008), a performance metric that also was used in Wilkins et al. (2016a). Results are presented in section three: First the sensitivity of inversion



calculations on input data uncertainty is demonstrated; secondly, the robustness of the calculated source term is tested by simulating a real case, where increasing amounts of satellite data are used, and modelled ash clouds are compared to observed ones. Discussion and conclusions are given in section four and five respectively.

## 2 Methods

5 ### 2.1 Source estimate calculations

Assimilated volcanic source estimates are calculated in this study by an inversion algorithm, based on the work given in Seibert (2000), and further developed to calculate the vertical distribution of volcanic emissions by Eckhardt et al. (2008) and Kristiansen et al. (2010). Stohl et al. (2011) presents modifications to the method to also produce time resolved emission estimates of ash for the 2010 Eyjafjallajökull eruption. Since the inversion method for volcanic ash emission estimates has 10 been extensively described in previous studies, further detailed description of the inversion method will not be given here, but some aspects are presented for the use in a forecasting setup.

The algorithm calculates an assimilated emission estimate using input data from a dispersion model and satellite retrievals, as well as a priori emission estimates. First, source receptor model data, representing all possible dispersion scenarios of the ash cloud, are matched with satellite data. For each grid point in the considered domain, modelled column loadings over 15 every hour of the assimilation time that exceed a certain threshold (here $10^{-12}$ g m$^{-2}$) emitted from a unit ash emission (1 kg s$^{-1}$ m$^{-1}$) released from one particular emission time and height are matched with the corresponding assimilation time and grid point of the satellite ash mass loading retrieval. Using a threshold exceedance criterion for the model data helps reduce the data volume and inversion cpu time. Model source receptor calculations are done by using an unit emission that are later scaled by the a priori emissions in the algorithm, making it possible to change the a priori estimate without performing new 20 model calculations. Model simulations used for the inversion are further described in section 2.2. Since grid boxes are used only where model results have ash loads above the threshold, the chance of the result being influenced by possible false positive ash retrievals in the satellite data, described in section 2.3, is reduced. On the other side, grid points which are unclassified, meaning it is uncertain whether they contain ash or not, are excluded from the inversion calculations. To reduce the amount of data and computational time, a randomly selected 70 % of the gridded data points, which hold satellite data 25 with definitely no ash, are discarded, similar to Stohl et al. (2011).

### 2.2 Model Simulations

Volcanic ash dispersion calculations are done with the EMEP (The European Monitoring and Evaluation Programme) MSC-W (Meteorological Synthesizing Centre - West) model described in Simpson et al. (2012), updates are in addition presented in the yearly EMEP reports (EMEP MSC-W, 2016). Model modifications to improve the description of ash dispersion such 30 as gravitational settling in all model layers, are described in Steensen et al. (2017). This new version of the model is called the emergency EMEP (eEMEP) model. Simulations are done with 3 hourly meteorological input from the ECMWF



(European Centre for Medium-Range Weather Forecast) IFS (Integrated Forecasting System) model with a horizontal resolution of 0.25 x 0.25 degrees in latitude and longitude, with 42 layers in the vertical. The model domain spans from 40 degrees to 80 degrees north, and 40 degrees west to 30 degrees east. The ash emissions are distributed over nine ash particle size bins from 4 µm to 25 µm particle diameter with an ash density of 2500 kg m$^{-3}$.

To produce source receptor model input for the inversion calculations, a unit amount of ash is released from 19 height intervals above the volcano as a pulse over a period of three hours. The three hourly ash emissions are distributed over the appropriate model layers given by the height intervals in the source emissions. Simulations are started every three hours until the whole period of interest is covered. With the current setup, ash is assumed to have a maximum residence time in the domain of 6 days before it is transported out of the domain or settled to the ground. The simulations therefore last for 6 days

after the pulse emission is released.

There are uncertainties connected to the model simulation caused by uncertainties in the meteorological input and assumptions about ash in the model. Stohl et al. (2011) tested the sensitivity of different model ash size distributions on the inversion calculations and found that, as the satellite observations only see a small range of ash size classes, changing the distribution over the size bins gave a negligible difference. The model simulations used as input to the inversion are also

done for an early part of the forecasted meteorological data when numerical weather prediction model uncertainties are still small. Errors caused by uncertainty in the meteorology and modelled size distribution are assumed minimal in our set-up, compared to the uncertainties connected to a priori emissions and satellite data and will not be studied here.

Figure 1 shows the timeline of the inversion calculation and the forecast via the eEMEP model simulation, both as used in this study and in the case of a real volcanic eruption. The a posteriori emission estimate calculated from the inversion routine

is used as the emission source term in the model simulations and can reach back up to six days counted from the forecast start time. An emission estimate for the forecast period is normally calculated as the average of the last 12 hours of the a posteriori source term. For practical reasons, the two model simulations (inversion method and forecast) are run separately from each other.

## 2.3 Satellite data

Ash satellite detection and retrievals are made using infrared measurements by SEVIRI on board the Meteosat Second Generation (MSG-2) satellite. MSG-2 is geostationary, centred at approximately 0 degrees latitude and has a 70 degrees view coverage (Schmetz et al., 2002). Pixel resolution is 3 x 3 km at nadir, while at the edge of the coverage it increases to 10 x 10 km. Observations are available every 15 minutes. Pixels are identified as containing ash if the brightness temperature difference (BTD) between the SEVIRI 10.8 µm and 12.0 µm channels (Prata, 1989) is below a certain threshold value, here -

0.5 K. The BTDs have been adjusted for water vapour absorption using the approach of Yu et al. (2002). Ash clouds give negative BTDs, ice give positive BTDs, while BTDs of water clouds are closer to zero.

For the inversion, satellite observations for every hour are used as input and interpolated by forward mean to the 0.25 x 0.25 degree model domain, two examples for April and May are shown in Figure 2. Grey areas in the plots represent unclassified





pixels where the satellite ash detection cannot determine if ash is present or not, that is, the BTD is around zero and pixels can therefore contain water, ice and ash. The ash detection can falsely classify ash in regions where there is no ash over land due to spectral land surface emissivity and for pixels with large viewing angles close to the edge of the SEVIRI coverage (Prata and Prata 2012). For the first date shown at the beginning of the eruption (15 April 2010 12 UTC, left plot Figure 2),

stationary ash clouds are detected both to the north and west of Iceland, while the main ash emission is transported east towards Norway, indicating that these ash clouds to the north and west are likely false positives. Other false positives are observed over Great Britain and in the North Atlantic Ocean for this time. For the second retrieval shown (7 May 12 UTC, right plot Figure 2), a large ash cloud is detected to the south west of Iceland that probably does not originate from volcanic emissions according to our understanding of the transport conditions. Because of the different thresholds and method used to

detect ash this cloud is not detected in the Francis et al (2012) and Wilkins et al (2016a) studies. False positives may be included in the inversion calculation because in a forecasting environment manual adjustments to the satellite data for these pixels can be difficult to accomplish, however, since model data where no ash is transported is disregarded, the chances of false positives being used in the inversion calculations are minimal.

The ash mass loading and effective ash particle radius are retrieved as described in Kylling et al. (2015). The retrieval is

based on a modification of the Bayesian optimal estimation technique used by Francis et al. (2012). There are several factors that affect the ash retrieval causing uncertainties in the calculated column loadings. Corradini et al (2008) studied uncertainties due to $\pm$ 2 K surface temperature and $\pm$ 2 % surface emissivity changes and found total mass retrieval errors of 30 % and 10 %, respectively. The same study also estimated a retrieval error of 10 % caused by variations in ash plume altitude and cloud thickness, and shows an almost approximately proportional uncertainty retrieval error due to water vapour.

Changing the ash type (e.g from andesite to the ash type from Volz (1973)) also give uncertainties in the total mass (Corradini et al., 2008, Francis et al., 2012, Wen and Rose, 1994). Wen and Rose (1994) studied the volcanic eruption at Crater Peak, Alaska in 1992 and found that total mass is doubled due to changes in ash particle size distribution. Kylling et al. (2014) found 30 % difference in total mass due to the assumed ash particle shape. The effect of meteorological clouds is seen to both increase and decrease the retrieved ash-mass loading (Kylling et al., 2015).

We assume andesite ash with refractive index from Pollack et al. (1973), spherical ash particles and a lognormal size distribution. The lognormal size distribution is described by the geometric mean radius and the geometric standard deviation. The geometric mean radius is related to the effective radius which is retrieved. To test the sensitivity to the shape of the size distribution the geometric standard deviation was varied between 1.5, 1.75, 2.0 and 2.25, which is a subset of the values used by Francis et al. (2012). The four satellite retrievals with different geometric standard deviations are henceforth referenced

as sat 1.5, sat 1.75, sat 2.0 and sat 2.25. Figure 3 shows the total ash mass in the domain for every hour during the Eyjafjallajökull eruption from the four satellite data sets. A larger geometric standard deviation gives a wider size distribution that includes more of the larger ash particles and therefore increased retrieved ash mass loading. The difference between the four satellite sets (fig. 3) show the effect the size distribution shape has on the observed ash loads. For the inversion algorithm an additional uncertainty is assigned to the ash loads in the grid cell. To see the effect of these other than



size dependent uncertainties on the inversion calculations, four uncertainties are assigned to the satellite data in separate inversion calculations; 0 %, 50 %, 100 % and 200% as a percent of the retrieved column load in each grid cell.

## 2.4 A priori emissions

Mastin et al. (2009) presents an empirical relationship between observed height and mass emission rate (MER) based on historic volcanic emissions.

$$MER = \left(\frac{H}{2.0}\right)^{-0.241} \times \rho$$

The observed plume heights (H) used in this study are given in Arason et al. (2011) with a three hour temporal resolution, density ($\rho$) for ash is equal as in the model simulations (2500 kg m$^{-3}$). A priori MER over the eruption period is shown in Figure 3. The a priori emission is distributed uniformly over the total emission column. Mastin et al. (2009) also gave a fine ash fraction for classified volcanoes over the globe based on previous eruptions. Larger tephra are assumed to fall close to the volcano and this tephra associated fraction of the total MER is not available for long range transport and is not included in our simulations. Large tephra is also not observed by the infrared satellite instruments using the BTD technique. Fine ash fraction for the Eyjafjallajökull volcano, classified as a silicic standard case is 0.4 which is higher than the 0.1 fine ash fraction used in Stohl et al. (2011) and Kristiansen et al. (2012). However, 0.4 are chosen to simulate a real case forecasting mode, where this fraction must be assumed as it is likely to be the only information available in the first phase of an emergency. Note this higher fraction involves significantly higher a priori emissions than used by Stohl et al. (2011) and Kristiansen et al. (2012). Note also that the observed heights used to calculate the a priori emissions here are on some occasions lower compared to the more uncertain heights used in the previous mentioned studies as the Arason et al. (2011) heights were not available at the time of these studies. Since a rather conservative a priori method is used here that does not favour any release height over another, the uncertainty range, within which the a priori estimate may fall, is chosen to be for four test cases 25%, 50%, 75% and 100%. We assume that this is informative to understand how uncertainty in the a priori emitted mass weights into the inversion calculations.

## 2.5 SAL metric

To measure the performance of the model as more observations are added to the inversion algorithm for the source term calculations as well as its forecast ability, the SAL (Structure Amplitude Location) scores (Wernli et al., 2008) are computed and evaluated. The SAL method is an object based quality measure originally developed to evaluate quantitative precipitation forecast with observations, and later applied to air quality forecasts (Dacre, 2011). The same satellite retrieval as used in the inversion assimilation is used as the observation field. This gives the opportunity to study how the model simulations in the analysis and forecast period become more similar to the assimilated data. Objects are identified in the forecast and observations field where parameter values exceed a certain threshold. The equations used to calculate the S, A





and L components of the method are described in Wernli et al. (2008) and Wilkins et al. (2016a), only a short description will be given here.

As in Wilkins et al. (2016) a more conservative ash threshold value of 0.5 g m$^{-2}$ is chosen to identify objects for the satellite and model fields, even though the satellite detection threshold is considered to be about 0.2 gm$^{-2}$ (Prata and Prata 2012).

5 For the amplitude component, the average ash mass over the domain are calculated for the modelles and observed fields. A is the normalized difference between these two averages, and ranges between -2 to +2, with 0 being the perfect forecast. An A value of +1 indicates a model overestimation by a factor of 3, and values of 0.4 and 0.67 represent model overestimations of 1.5 and 2 respectively.

The structure component compares the normalized volume objects by scaling the ash loading with the maximum ash loading 10 within each object. Forecast and observed objects are then weighted proportionally to the ash mass of the objects. S is the normalized difference between these weighted modelled and observed volumes. S also ranges between -2 to +2. S is positive when the model ash field is too spread out and flat, while a negative value correspond to a model field that is peaked and/or too small.

The first part of the L component measures the normalized distance between the centres of mass for the modelled and 15 observed fields. Different ash clouds can have the same centre of mass, and the second part of L considers the averaged distance between the centre of mass of the total field and individual objects. Both parts of L ranges between 0 and 1, a maximum of L is +2. The definition of L is however insensitive to the rotation around the centre.

The combined SAL score is given by (|S|+|A|+L), a perfect forecast is given by 0, while the maximum score is 6. The possibility of a perfect score forecast for modelled fields with a posteriori emissions and satellite retrievals is minimal 20 because of the difficulties detecting ash in the satellite data, however the tendencies of a possible improvement in the forecast can be analysed by the use of this method.

The SAL scores are calculated for every 12 and 00 UTC time step after the start of the eruption in the April and May period for all the forecast and assimilation period. Two 48 hour forecast experiments are characterized, one with average and zero emissions estimate included in the forecast period. To only compare the ash clouds that are in areas where the model 25 calculations show ash levels above a (very low) threshold value (see above), false positives in the satellite data are not included. In addition areas with unclassified pixels in the satellite data are excluded for both the observed and modelled fields.

## 3 Results

### 3.1 Emission estimate uncertainties

30 Multiple inversion calculations are performed using the four satellite data sets with the different size distribution shape (sat 1.5, sat 1.75, sat 2.0 and sat 2.25) in combination with varying the uncertainties connected to the a priori source estimate (25%, 50%, 75% and 100%) and satellite retrieval uncertainty due to other factors than size distribution (0 %, 50 %, 100 %





and 200%.). Figure 4 shows the a priori emission estimate over time during the two periods in April and May, as well as the total range in the a posteriori emission resulting from the multiple inversion calculations. As the amount of ash emitted in the a priori is a function of the observed emission height at the volcano, more ash reflects a higher observed emission column. All our a posteriori estimates reduce the emissions from the a priori, suggesting that the default parameter value for the fine ash fraction of 0.4, as taken from Mastin et al. (2009) is indeed too high as discussed in section 2.4. Other parameters such as density and plume height may also result in too much a priori emission.

In April, a high emission column at the start of the period is followed by reduced column height observations before more ash is emitted again from 16 April 9 UTC. The a posteriori show a large range of solutions for the first plume released. During the low emission period in April all the a posteriori follow the a priori as the inversion can not constraint the a posteriori solution without any satellite observations. On 17 April when the satellite detected more ash, the a posteriori emissions are strongly reduced compared to the a priori estimate, similar to what is found in previous inversion studies using model input data from FLEXPART and NAME (Stohl et al., 2011; Kristiansen et al., 2012). The May period also starts with a high a priori emission estimate, followed by a period with almost constant lower a priori emissions. The a posteriori estimate is strongly reduced for the whole period.

Figure 5 shows the average vertical distribution in the emissions over the April period for the a priori and all inversions performed, grouped into eight ensembles. In Figure 5a) the a priori uncertainty is set to 75 %, and for each of the four different satellite data sets the other-than-size uncertainty of the satellite data is varied from 0 to 200%, giving the shown spread in the vertical emission distribution estimate. In figure 5b) the other-than-size satellite data uncertainty is set to 100% and the a priori emission uncertainty is varied from 25 to 100%. The resulting spread in vertical emission distribution for the different satellite data sets represent the a priori uncertainty.

All a posteriori estimates are strongly reduced compared to the a priori especially at altitudes below 4 km. The reduction of ash in the resulting emission estimate is proportional to the reduction in amount of ash in the satellite retrievals. A posteriori for the satellite data set with most ash (sat 2.25) have higher emissions than the other satellite data set with less ash. As the other-than-size satellite uncertainty is a percentage of the retrieved ash for each grid point, the satellite set with the highest column loads also shows the largest spread (Fig 5a). Comparing figures 5a) and 5b) this spread is however much smaller than that caused by varying the a priori uncertainty (Fig 5b).

Another feature can be found in these plots of the vertical distribution of the emissions and the spread in different heights. Since the inversion redistributes ash emissions to the heights where trough transport processes the best match to satellite observations, the vertical distribution is changed from a priori. The emission close to ground is reduced and the largest spread due to a priori uncertainty is at this altitude (below 4 km) (Fig 5b). More trust in the a priori estimate (low uncertainty) causes the a posteriori estimate to deviate less from the a priori emission profiles (right part of the result envelopes in Fig 5b). The most left profile representing the lowest emission term is attained with high uncertainty for the a priori emissions and little ash mass retrieved by the satellite (sat 1.5). Therefore the a posteriori estimates for this satellite





data set have the largest spread as a function of variation in a priori uncertainty. The corresponding vertical emission distribution plots for the May period show similar results (not shown).

The spread in a posteriori estimates caused by varying the inversion input, both with regards to the column loads in the satellite retrieval and uncertainties connected to them and the a priori emission uncertainty represent the ambiguity in the a

posteriori. Ideally, uncertainties should be set at values that are representative of the real uncertainties connected to the data, however these uncertainties are often not well known at the start of an eruption. Using a range of uncertainty values provides insight into the confidence of the results and should ideally be performed during real case operational setting. This is unfortunately computationally demanding and practically not feasible. The spread of a posteriori estimates presented here can however contribute to further interpretation of the a posteriori results in the case of future volcanic eruptions dealt with

in an operational setting.

For the remainder of the results presented in this study, an other-than-size satellite uncertainty of 100 % and an a priori uncertainty of 75 % will be used on the 1.75 satellite retrieval data that are termed "the reference a posteriori", shown as magenta line in Figure 4. The inversion result and associated simulation is typical for our ensemble. Table 1 shows for instance that the total emitted fine ash for the a priori emission estimate is reduced by around 45 % for April and 65 % for

May in the reference a posteriori seen against the a priori emission estimate. The different ranges of the total a posteriori ash emission for the different satellite retrievals, the other-than-size satellite uncertainties and the a priori uncertainties input are also calculated by fixing the other two parameters as the reference. For both periods, the largest spread is caused by the four different satellite retrievals while changing the other-than-size satellite uncertainty produces the smallest spread. Since this smaller spread is seen to depend on the amount of ash in the satellite retrieval, forecast simulations are therefore also done

for the 2.25 satellite retrieval with the same uncertainty estimates as for the reference (orange line in Figure 4).

### 3.2 Inversion in forecasting-mode

In a real volcanic alert case, more and more information will become available while the event is enfolding. To test and investigate the change in the a posteriori estimate as more observations become available, new inversion calculations are made every 12 hours of the 4 day periods in April and May. The first inversion calculations become available on 00 UTC 15

April and 00 UTC 6 May with observations accumulated up until that time (24 hours of satellite observations). It would have been possible to do the first inversion calculations before this first time step, the satellite observations often have problems detecting the ash close to Iceland due to the high optical thickness of the ash cloud close to the volcano so only a few satellite observations are available. Figure 6 and 7 show the a priori as well as a subset of the consecutive a posteriori vertical distribution emission estimates at three-hourly resolution, calculated with observations that would have been

available up until 00 UTC for each day of interest in April and May, respectively. Comparing multiple consecutive estimates illustrates how robust the a posteriori emission is, especially for the first high ash emissions in the periods.

The first a posteriori estimate calculated with satellite data up until 15 April 00 UTC shows a strong reduction in the emissions compared to the a priori over the 9 UTC to 18 UTC 14 April emission columns (figure 6). Adding another 24



hours of satellite observations increases the emissions at 8 km height, while reducing the emissions closer to ground. This redistribution is caused by the transport patterns seen in the satellite ash images, which imply that transport happened at high altitudes and not at low altitudes. Even more observations including days 3 and 4 only change the 14-15 April emission estimate slightly. Figure 4 shows that the a posteriori estimates have minimal differences compared to the a priori between

15 April 12 UTC and 17 April 00 UTC. The larger impact of the inversion on emission estimates altering the a priori to rather low values for the second part of the emissions on 17 April is caused by only a few hours of satellite observations.

Figure 7 shows that the high emission estimates during the first 24 hours of the May period overall are also reduced early on in the first inversion result. The first nine hours show agreement between a priori and a posteriori. For the 15 UTC to 21 UTC period on 5 May, there are numerous height levels where emissions are zero. This is caused by how the inversion

algorithm handles unphysical negative inversion calculations that are caused by inaccuracies in model and data. The standard error for these negative source vector elements are reduced and inversion calculations are repeated until the sum of all negative emissions is less than 1 % of the sum of positive emissions (Eckhardt et al. 2008). Small negative emissions that are still present in the estimate are set to zero. By adding more observations, these artefacts are reduced and the negative values are replaced by very low emissions, indicating a more confident estimate. Another noticeable factor is the number of

observations needed to reduce the emission released between 21 UTC 6 May and 00 UTC 7 May. The first estimate calculated at 8 May 00 UTC shows little reduction, only when more observations up to 8 May 12 UTC are included (not shown, but visible in the 9 May 00 UTC inversion) these emissions become small. Similar difficulties to correct the night-time emission are seen for the 21 UTC 7 May to 00 UTC 8 May emission as well as during the April period. The reason for this is beyond the scope of this study however the results indicate that there is an increased uncertainty connected to the

inversion method attempting to derive night-time emissions.

Figure 8 shows where the differences in vertical emission distribution are located when two satellite data sets are fed into the inversion calculation for the two periods. Although more ash in the satellite retrieval sat 2.25 causes the source emission to have higher emission fluxes, the change in a posteriori emissions when adding more observations are similar for the 1.75 and 2.25 satellite retrievals for both the April and May period. Although the biggest differences are seen for the same emission

time as the maximum flux of the emission estimate during the two periods, the largest differences are closer to ground. The increase of the maximum emission fluxes at the higher levels between the two satellite retrieval is minimal. During April, the highest emission level is transported quickly out of the domain while lower levels are transported over Europe with larger differences between the satellite retrievals. For the May period, the large difference below 4 km are caused by the satellite retrieval in sat 2.25 having an increase over time in column loading for the southerly part of the plume that is not present in

the sat 1.75. This different increase is ash loading over time between the different satellite retrievals is because the size distribution enters the radiative transfer equation non-linearly. The emissions released at higher levels have been transported further north and are not affected by this.





### 3.3 Forecast model results compared to satellite observations

Figure 9 show the ash distribution satellite retrievals (sat 1.75) every 12 hour from 16 April 12 UTC to 17 April 12 UTC. It also shows corresponding model results for a simulation with an emission estimate calculated up to the satellite observation and corresponding results including a 36 hour forecast period, applying a mean forecasted emission term established for the

12h preceding to the start of the forecast. All the model results have more extensive ash clouds compared to the observed ash clouds. Maximum concentrations are however high in the observed data (10.5 g m$^{-2}$, 9.5 g m$^{-2}$ and 6 g m$^{-2}$ on 16 April 12 UTC, 17 April 00 UTC and 17 April 12 UTC respectively). Disregarding areas close to the volcano, Figure 9b shows that initially simulated ash concentrations, right after the assimilation period, have the highest concentrations of ash in the area of observed ash, but with a maximum of 5.1 g m$^{-2}$ the modelled ash column values are lower compared to the maximum values

in the observations. The forecast started using the first emission estimate, covering emissions released before 15 April 00 UTC (Fig. 9c), have high a posteriori emissions and therefore also high forecast emissions causing a large amount of ash to be released into the atmosphere, and have maximum column load of 19.5 g m$^{-2}$ in the area where the satellite retrieve ash. The model simulation does not manage to transport narrow ash clouds with high concentrations due to numerical diffusion and the optimized field (Fig 9b) therefore have smaller maximum values. For the next forecasts starting 12 hours later (Fig.

9f), the emissions are already reduced. Differences between the forecast starting on 17 April 00 UTC (Fig. 9e) and the 36 hour forecast (fig. 9f) are minimal due to low emissions during this time, both have maximums over central Europe at 4.0 g m$^{-2}$ and 5.1 g m$^{-2}$ for the initial and forecast respectively. In both model simulations there is an area with higher column loads to the south of Iceland due to more emissions being accumulated by weak northerly winds. No ash is retrieved in the satellite observation. For the satellite plot in Fig. 9g retrieved 12 hours later, ash is detected to the east of Iceland that is released

before 12 hours prior demonstrating the difficulty of retrieving the opaque ash clouds close to Iceland. For this retrieval, there is also no ash detected over Europe, even though ash was observed over Europe at this time (Pappalardo et al. 2013). The exemplary results in Figure 9 show that for a 36 hour forecast, being a long forecast including an unknown emission estimate, rapid changes in the mass eruption rate may lead to significant error.

While the ash observations during the April episode are characterized by small observed ash clouds with high ash

concentrations, the observations of the ash during the May period show larger ash clouds with lower column loadings. Figure 10a and 10b show retrieved satellite ash on 8 May 12 UTC for the 1.75 and 2.25 size distributions. Model results with emission estimate calculated with the respective satellite retrieval calculated up to 8 May 12 UTC and a 36 hour forecast from 7 May 00 UTC is also shown. Because of small ash emission estimated from 6 May 00 UTC onwards the differences between the forecast emission estimate and the assimilated estimate is minor, except for more ash south of Iceland for both

satellite retrievals for the initial simulation. As discussed in the previous section, for the most southerly ash cloud in the 2.25 satellite retrieval the ash column loads increase over time and cause the cloud in the model results in Figure 10e to have more ash than in the forecast simulation (Fig. 10f) even though this emission is already inverted from previously





observations of the ash cloud. This change in the emission estimate for distant, early emissions caused by more satellite observations demonstrates the ability to improve ash simulations, if ash was obscured by clouds in earlier retrievals.

## 3.4 Performance quantification forecasts

The SAL score and its components (see section 2.5) are calculated every 12 hours during the simulation periods including
the assimilation period plus a 48 hour forecast to quantify the performance of the model as more and more observations are added. SAL scores are also calculated for a simulation using the a priori estimate to estimate how the assimilated source term improves over the a priori.

For the April period, the retrieved satellite ash clouds are small compared to model clouds and consequently the S and A scores become very high. An exception is for the 17 April 00 UTC retrieval where the areas with unclassified retrievals are
large over Europe (Fig. 9d). This large unidentified area is due to high emissivity over land during night time that disrupts the brightness temperature retrieval quality. Removal of these areas in the model data causes the fields to be more comparable. Even though the model ash clouds are indeed larger and more spread than the observed ash for the period, comparing the observed and modelled fields for this time provides some information about how the amount of ash are changed by adding more observations. Table 2 shows the SAL scores for the two satellite retrievals (sat 1.75 and sat 2.25)
and the corresponding model stimulations with the emission estimates constrained by the satellite retrievals. For the simulations where the assimilation period and inversion estimate ends before the comparison time (15 April 00 UTC to 16 April 12 UTC) the 12h averaged forecast emission estimate is added, while for the rest of the model simulations (17 April 00 UTC to 18 April 00 UTC) the observation is included in the assimilation calcualtions. Compared to the a priori estimate, all forecast model results are worse for the structure (S) component because of the too spread out model fields. The amplitude
(A) scores that measures the amount of ash in the domain is however improved for the second assimilation with forecast estimate (0415 12 UTC + 36 hours) and the preceding simulations.. The structure score does not improve until the 17 April 00 UTC satellite observation is included in the assimilation (three lasts lines in Table 2). This improvement is due to a smaller area over the 0.5 threshold over Europe in these simulations.

Figure 11 shows all the SAL scores in the May period for satellite observations and the model simulations for the sat 1.75 (a)
and the sat 2.25 (b) size assumptions. SAL score calculations are in addition done for a 48 hours forecast with the last 12 hours average emission estimate (dashed lines) and zero emission (solid lines) over the forecast period. Because of the optical thick ash cloud close to the volcano there is no ash originating from the Eyjafjallajökull eruption in the 5 May 12 UTC satellite retrieval, it is not possible to calculate the location (L) and structure (S) scores for these times and the amplitude (A) gives the worst score (2) due to infinitely more model ash than satellite. SAL scores generally are better
during the May period because of the increased amount and areas with retrieved ash for the observation field compared to the April period. The first emissions in the May a posteriori estimate are not reduced enough in the inversion calculations causing the A score to be high for the May 6 00 UTC comparison in all the model comparisons. Transport later in the period aligns this model ash released early with ash released later in the period forming the southern ash cloud (Fig. 10). For the 7





May 00 UTC comparison time, the two forecast estimates show good results for the S score, while the model simulations with this observation time late in the assimilation period performs worse. Further into the period as the observations time becomes earlier in the assimilation period the model performs better for both A and S. The two model simulations with assimilation period up to 9 May 00 UTC score better for the A and S than all the other model simulations for most of the

comparison times, even though the emission estimate did not change much during this time.

The A and S scores are positive for most comparison times showing that the model fields have more ash and the fields are more spread out than the satellite observations. This can be explained by the difficulty of retrieving ash close to the volcano and ash that are obscured by meteorological clouds.

Ash locations score (L) between satellite and model data is low, both because of the centre of mass is close to each other in

the domain, and in addition the L score for the idealized fields shown in Wernli et al. (2008) are lower than the S and A values. Low L values also indicate that the transport of ash in the model compare well to observations which also indicate that the ash emissions are placed in the right layer.

Although the emission estimates are calculated by using different satellite retrievals and compared to their respective satellite data, the scores for the two satellite data sets do not show large differences. The difference in the S score on 17 April 00

UTC is caused by less ash in the small objects for the 1.75 observed fields. For the May period, the S scores are similar to each other however more ash in the 2.25 satellite retrievals compare better to the amount of ash in the model simulations leading to a better A score for the 2.25 satellite retrievals.

## 4 Discussion

Emission fluxes in the a posteriori estimates depend on the amount of ash in the satellite retrieval and the weighting of

uncertainties connected to the input data to the inversion. Giving the a priori estimate a high uncertainty causes the a posteriori estimate to deviate from the a priori while assigning a high uncertainty to the satellite data forces an inversion solution closer to the a priori emission term. It is therefore important that the a priori and satellite uncertainties connected to these values represent reasonable assumptions. Default settings for an operational setup can ignore some aspects of a volcanic eruption for tephra size distribution and amount in emission.

Of major importance is the uncertainty in the satellite data input to the inversion, and especially the change in ash loads by using different assumptions about the shape of the ash size distribution. The results in this study show that the spread in a posteriori estimates due to other-than-size satellite uncertainties are much smaller compared to the spread when using the four different satellite sets with different size distributions. For the a posteriori estimate it is therefore important to use the best available assumptions in the satellite retrieval rather than correct other-than-size uncertainty assumptions.

Satellite retrievals from other satellites instruments with better spatial resolution such as for example MODIS (Moderate Resolution Imaging Spectroradiometer), IASI (Infrared Atmospheric Sounding Interferometer) and VIIRS (Visible Infrared Imaging Radiometer Suite) may provide more confidence in the extent of the ash clouds (Clarisse et al., 2010). Such



retrievals may carry similar uncertainty for finding ash mass, but may bring additional size info and separation of ash from cloud. Satellite information can also give information about the height of the ash layer. This may be obtained from dual view instruments such as SLSTR (The Sea and Land Surface Temperature Radiometer), and space borne lidars such as CALIOP (Cloud-Aerosol Lidar with Orthogonal Polarization) also provide valuable information if their narrow footprint match the ash cloud (Winker et al., 2012).

The SEVIRI satellite observations have high temporal resolution as a new retrieval is available every 15 minutes for the whole domain. Polar-orbiting satellites on the other hand, may only observe a small part of the domain during an overpass. Stohl et al. (2011) show that performing inversion with only IASI retrievals may provide a too small sampling size to constrain the solution. Ash mass loadings from other satellite retrievals with better aerosol detection capability are nevertheless useful for comparisons with the amount of ash in the SEVIRI retrieval and a possible combination of the satellite retrievals with the SEVIRI retrieval for the inversion.

The a posteriori solution is found to only use the a priori estimate in the absence of ash in the satellite retrievals, this solution is independent on the uncertainty settings for a priori and satellite data. A good a priori estimate is therefore important for these cases. Observed heights from Arason et al. (2011) obtained by weather radars are used in this study, and the heights show a good match with the maximum a posteriori heights. However the fine ash fraction is found to be too large causing too much ash to be released during the April period. Observations and more information are needed to produce a good a priori estimate. At the time of the Eyjafjallajökull eruption, Iceland had only one operational weather radar to observe the plume height, situated at Keflavik International Airport, 155 km to the west of the volcano (Arason et al. 2011). Another permanent weather radar is now situated on the eastern part of Iceland, and two mobile radars are prepared (Jordan et al. 2013). Monitoring of activity on Iceland is also improved by the FUTUREVOLC project (http://futurevolc.hi.is), and will increase the amount of observations available in the case of future volcanic eruption.

Even when using higher a priori emission heights for the estimate for the Eyjafjallajökull eruption as in Stohl et al. (2011) and Kristiansen et al. (2012), their results show that the inversion algorithm places ash at equal heights as found in this study. The fine ash fraction of 0.1 used in Stohl et al (2011) and Kristiansen et al. (2012) gives however a better match than the too high 0.4 used in this study for the periods where the satellite observations are too few to constrain the a posteriori ( and a posteriori therefore only use the a priori estimate). Eckhardt et al. (2008) showed that a posteriori estimates calculated with no emission in the a priori emission gave similar results to a posteriori estimates calculated with estimated emissions in the a priori. A posteriori estimates are also calculated with low ash emissions in the a priori estimates in the Moxnes et al. (2014) and Kristiansen et al. (2015) studies. Ash can however be obscured by meteorological clouds and optical thick ash clouds may not be detected, so an a priori emission with ash is therefore considered more conservative. A parallel sensitivity calculation with no or little ash in the a priori estimation is possible in case of a volcanic eruption but not done in this study.

The insertion method presented in Wilkins et al. (2016a) and a refined method in Wilkins et al. (2016b) only takes into account the ash in the satellite retrievals and adds no additional emissions from the volcano in the forecast, eliminating the concerns with the a priori emissions for periods with no ash detected. By inserting several ash retrievals in the model field



over several times, possible undetected ash can be included in the calculations as it may become visible in later satellite retrievals. Comparing the insertion and the inversion methods for 16 April 2010 12 UTC show that the insertion method have ash clouds only at similar location as the observations while the results presented here have too extensive ash clouds. Wilkins et al. (2016a) also present SAL metric results from 8 May 2010 9 UTC. Although not calculated at the same satellite

retrieval time, the SAL metric results in this study for May are better for a long forecast period. The amplitude score for the insertion method show that the averaged mass in the model results is less than retrieved ash, while in this study model simulations have more ash than in the retrieval. Some of these differences are caused by the inversion calculations using only the a priori estimate in the absence of satellite observations, for the April period, and how the observations field are defined for the SAL score calculations. In Wilkins et al. (2016a) the observed satellite data is represented by the maximum values

retrieved over the previous hour, while in this study, the observations are strictly the ash loading retrieved at the time studied. Another reason is caused by the difficulty the satellite retrievals have detecting high density ash close to the volcano leading both to too much ash in this study that includes these ash clouds in the forecast and possibly too little ash in the insertion method that does not use emissions over the forecast period.

## 5 Summary and conclusions

In this paper the inversion method is tested in an operational forecasting setting over two short periods of four days during the Eyjafjallajökull eruption. Both of these periods started with high ash emissions during the first day and while the observations of ash during the April period indicated small clouds with high column loadings, the retrieved ash clouds during the May periods were larger in extent with lower column loads. This provides an opportunity to explore the feasibility of using an inversion method to constrain emission in an operational setting where the impact of volcanic eruptions on air

traffic shall be assessed. The observed ash cloud during the April period are shown difficult to simulate in the model due to diffusion and the model results with the a posteriori therefore have ash clouds that are more spread out and ash column loads that are lower compared to satellite. The ash clouds observed in the May period are better simulated by the model.

A posteriori emission estimates are calculated with the inversion algorithm for four different satellite data sets with different spread in size assumptions that affect the retrieved ash column loadings. Note that the satellite data also contain areas with

unclassified pixels where the satellite retrieval is not able to decide whether ash is present or not. These areas are ignored by the inversion algorithm. The effect of different uncertainties connected to the input satellite data and a priori estimate in the inversion are studied and multiple inversion calculations are documented. Because of the high fine ash fraction (0.4) assumed for Eyjafjallajökull as a silicic standard volcano (Mastin et al. 2009), the a priori estimate has too emission and all the calculated a posteriori emissions are reduced by the inversion. Inversion calculations for the four satellite retrievals with

the least ash have the highest deviation from the a priori, and changing the uncertainties connected to the a priori term leads to large spread in the a posteriori estimates. Other-than-size uncertainties connected to the satellite retrieval are found to have lower effect.



As the inversion routine forces the source term and the model simulations to be more similar to the observed ash values, ultimately better quality data are needed for the retrieved column load values. Combining and comparing the SEVIRI satellite data with ash retrieval from other satellite instruments with different spatial and temporal resolution and different viewing angles are therefore necessary.

In a forecasting mode, the change in a posteriori estimates by adding more observations every 12 hours show, that although the a priori emissions are too high they are reduced early on with only a small amount of satellite observations. Adding more observations at later times of the ash cloud, further away from Iceland, causes the inversion to redistribute the ash emissions to higher altitudes in the Eyjafjallajökull case. The redistribution is caused by ash originating from these upper level emission heights witch are found to match better with the location of the observed ash. The results show that the

change in a posteriori estimate by adding more observations are minimal after 36 to 48 hours, in particular for those times where high ash occur. Emission times with no significant ash emissions are reduced after only a few satellite observations, exceptions are found for the night-time emission estimate between 21 and 00 UTC. During the April period, large ash emissions were followed by a period of no or insignificant ash emissions, where no ash is detected in the satellite retrieval. As the a posteriori estimate uses only the a priori for emission times that are not matched with satellite observations, more

information about the source term are necessary. For future Icelandic volcanic emissions such information will be available due to the increase in radar coverage in Iceland since the Eyjafjallajökull eruption.

The SAL scores show that the model at most times have more ash that is more spread out than the observations. Discrepancies between the observations and model results are explained by too much ash in the a priori, and undetected ash in the satellite retrieval close to the volcano or obscured by meteorological clouds. Model results with a posteriori emissions

decreases the ambiguity to both the forecast and the satellite observations by obtaining model ash loads more comparable to satellite values, and improving confidence in the satellite data by identifying areas with false positives and possible undetected ash.

### Acknowledgements

The work done for this paper is funded by the Norwegian ash project financed by the Norwegian Ministry of Transport and

Communications and AVINOR. Model and support is also appreciated through the Cooperative Programme for Monitoring and Evaluation of the Long-range Transmission of Air Pollutants in Europe (No: ECE/ENV/2001/003). This work has also received support from the Research Council of Norway (Programme for Supercomputing) through CPU time granted at the super computers at NTNU in Trondheim.





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





**Table 1: Total fine ash emissions in Tg over the April and May period for the a priori estimate and the reference a posteriori with the satellite retrieval with 1.75 geometric standard deviation, 100 % uncertainty in the satellite data and 75 % uncertainty of the a priori emission. The minimum and maximum a posteriori emission with varying the satellite input data with different retrieval assumptions, uncertainty connected to the satellite retrieval and a priori uncertainty while keeping the other uncertainties equal to the reference. The percent the sensitivity spreads are on the reference are also calculated for the two periods.**

|  | April | May | % of reference April | % of reference May |
|---|---|---|---|---|
| A priori | 17.4 | 13.3 |  |  |
| Reference a posteriori | 9.5 | 4.7 |  |  |
| Sat ret. (min/max) | 9.4/11.0. | 4.2/6.4 | 26% | 47% |
| Sat uncert (min/max) | 9.4/9.5 | 4.7/4.8 | 1% | 2% |
| A pri uncert (min/max) | 9.0/11.4 | 4.7/5.8 | 25% | 23% |

.

**Table 2: Structure Amplitude Location (SAL) scores (ranging from -2 to 2 for structure and amplitude, and 0 to 2 for location, best is 0 for all) for different model simulations for comparison on the 17 April 00 UTC using satellite retrievals (sat 1.75 and sat 2.25, see text). The model simulations that end the assimilation window before the comparison, and then use assumed forecast emission are marked as +hh hours. The last three lines correspond to simulations where the forecast starts after the observation comparison time.**

| Model forecast | Structure | | Amplitude | | Location | |
|---|---|---|---|---|---|---|
|  | sat 1.75 | sat 2.25 | sat 1.75 | sat 2.25 | sat 1.75 | sat 2.25 |
| A priori | 0.20 | 0.37 | 1.82 | 1.69 | 0.26 | 0.22 |
| Forecast starting before 17 April 00 UTC | | | | | | |
| 0415 00 UTC + 48 hours | 0.92 | 1.05 | 1.90 | 1.83 | 0.18 | 0.13 |
| 0415 12 UTC + 36 hours | 0.57 | 0.77 | 1.75 | 1.62 | 0.20 | 0.16 |
| 0416 00 UTC + 24 hours | 1.00 | 1.23 | 1.21 | 0.91 | 0.32 | 0.25 |
| 0416 12 UTC + 12 hours | 0.36 | 0.75 | 1.66 | 1.47 | 0.24 | 0.22 |
| Simulations with observation included in the assimilation to the inversions | | | | | | |
| 0417 00 UTC | -0.21 | 0.32 | 1.79 | 1.66 | 0.23 | 0.21 |
| 0417 12 UTC | -0.18 | 0.35 | 1.78 | 1.65 | 0.23 | 0.21 |
| 0218 00 UTC | -0.15 | 0.42 | 1.78 | 1.65 | 0.24 | 0.21 |



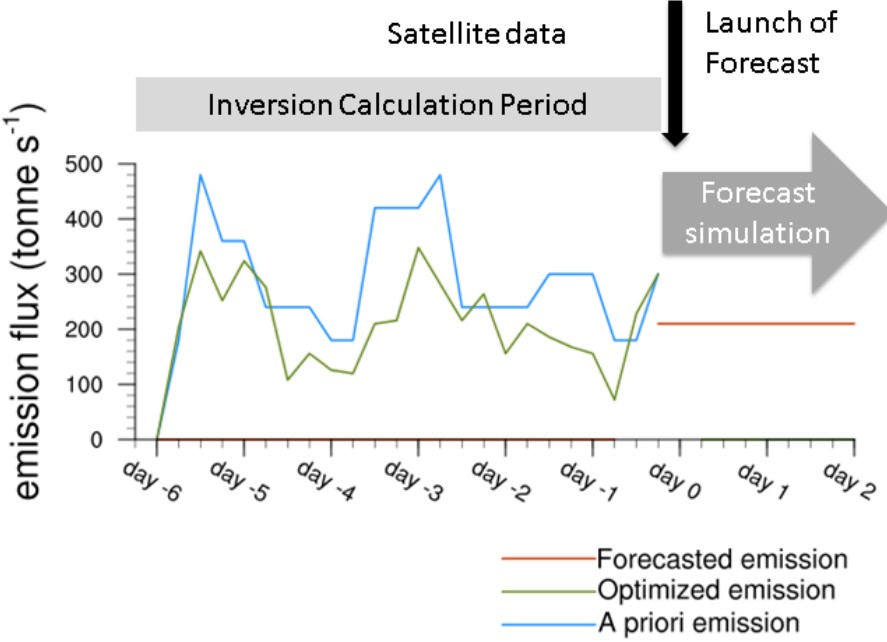

**Figure 1: Scheme of how the evolution of ash emissions used in the eEMEP model simulations may look like, with an a priori emission estimate, the calculated a posterior (optimized) emission estimate and a forecast emission estimate.**

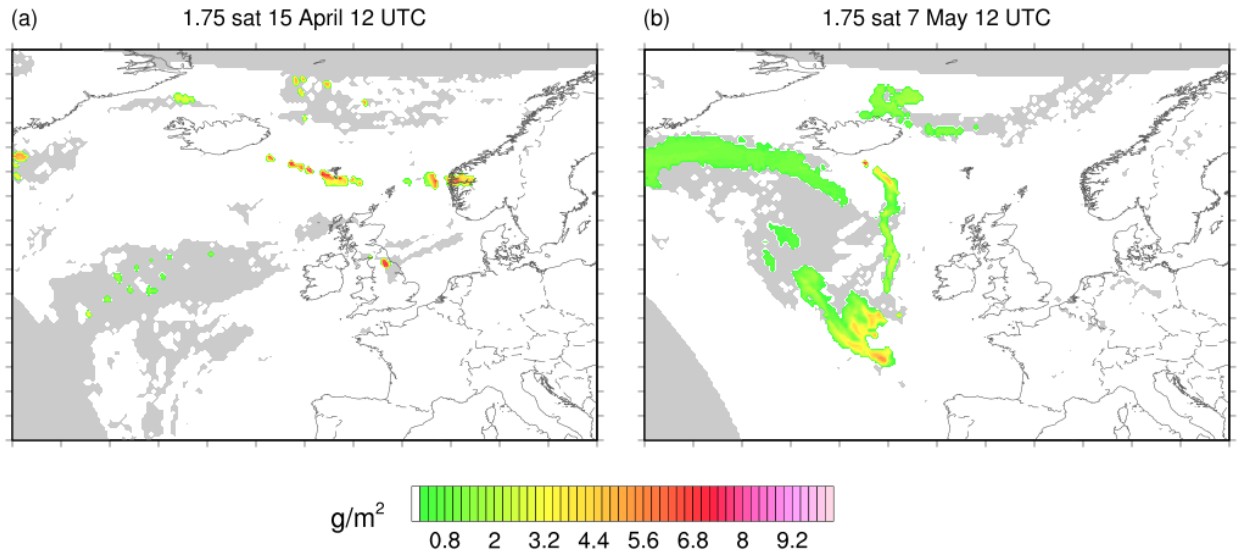

5    **Figure 2: SEVIRI satellite ash mass loading with a 1.75 lognormal size distribution on 15 April 12 UTC and 7 May 12 UTC 2010. The grey areas show the unidentified pixels, where the ash retrieval can not distinguish if contain ash or not.**





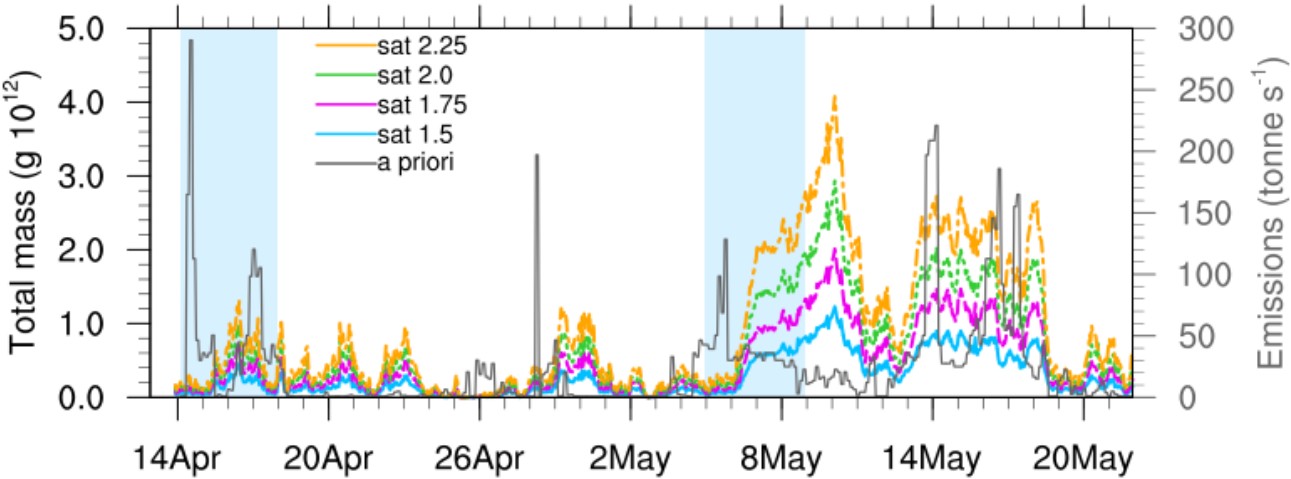

**Figure 3: Left axis, the total mass of ash in the domain for the four satellite retrievals with different size distribution assumptions (sat 1.5 – 2.25), for every hour over the entire Eyjafjallajökull eruption period. Right axis shows the emissions in the a priori estimate calculated from observed plume height at the volcano. The blue shaded areas indicate the periods studied in the paper.**

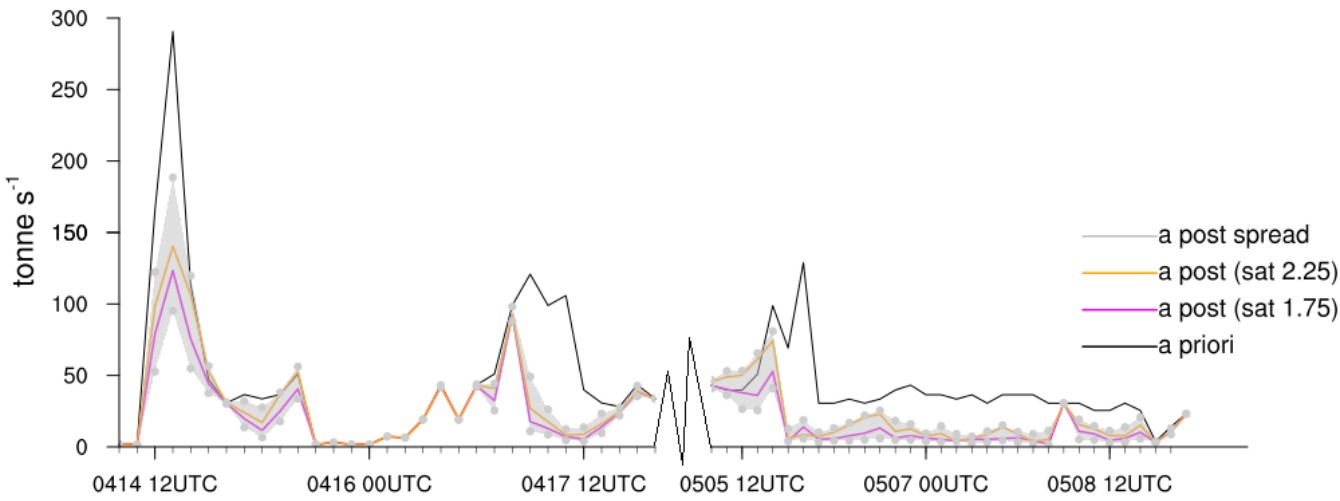

**Figure 4: A priori ash emissions and the spread of a posteriori ash emissions calculated by the inversion algorithm using the different uncertainties and satellite data sets during the April and May periods (the break on the x-axis indicate the change in time**
10 **periods). Magenta and orange lines are a posteriori emissions calculated from inversions assuming a priori uncertainty of 75% and satellite uncertainty set at 100%, using a spread in ash distribution of 1.75 and 2.25 in the satellite retrieval.**




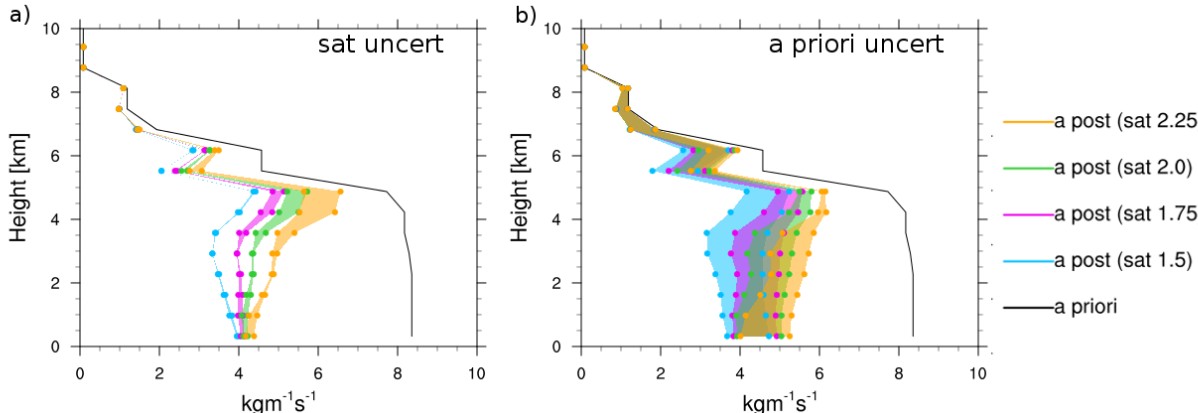

**Figure 5: Spread of a posteriori for the four satellite data sets with the four different size distribution assumptions (sat 1.5 – 2.25). Left plot show the spread in a posteriori caused by varying the uncertainty connected to the satellite data, with a priori uncertainty set at 75 %. The right plot shows the spread in a posteriori caused by varying a priori uncertainty, with a constant satellite uncertainty at 100 %.**







**Figure 6: Vertical emission distributions over the volcano with three hour resolution, given in kgm⁻¹s⁻¹. A priori source term (top row) and a posteriori source terms (middle and bottom row) by using satellite observations up until the start of the forecast time (vertical black line) over the April period. Only the a posteriori term for the 00 UTC forecasts are shown.**






**Figure 7: Same as figure 6 but for the May period.**





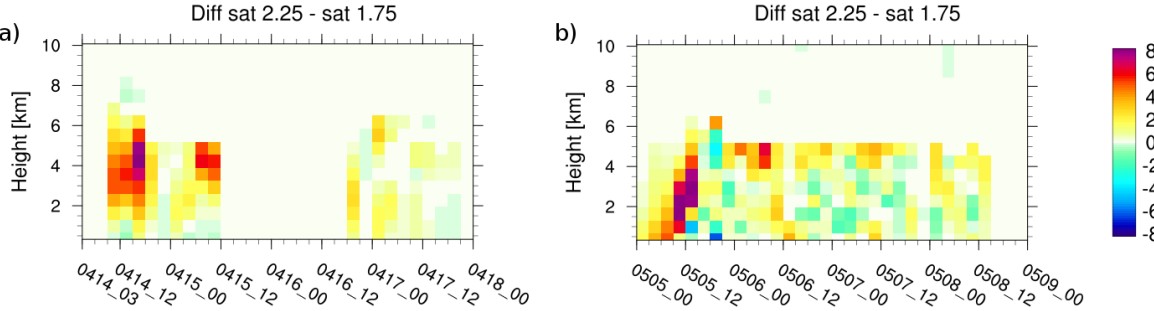

**Figure 8: The difference in emissions (kg m⁻¹ s⁻¹) between the a posteriori estimates for the inversions using the 2.25 and 1.75 satellite data sets over the two periods in April (left) and May (right).**




**Figure 9:** Left column shows SEVIRI satellite retrievals assuming the sat 1.75 size distribution at 16 April 12 UTC (a), 17 April 00 UTC (d) and 17 April 12 UTC (g). Blue areas indicate where the satellite has false positives detection of ash. Middle column shows model simulations with a posteriori emissions calculated from the inversion assuming the 1.75 size distribution, using all satellite retrieval up until the time indicated above the figure (same as satellite). The right column shows model forecasts for the same time as the two first columns, but with a posteriori emissions calculated with satellite observations up to 36 hours before and a forecast emissions term for the remaining 36 hours. The green line encircles objects used for SAL (Structure Amplitude Location) scoring, where ash exceeds 0.5 g m$^{-2}$ limit for model an observed ash. Ash released in the forecast term is shown with a dashed line (only the rightmost column).





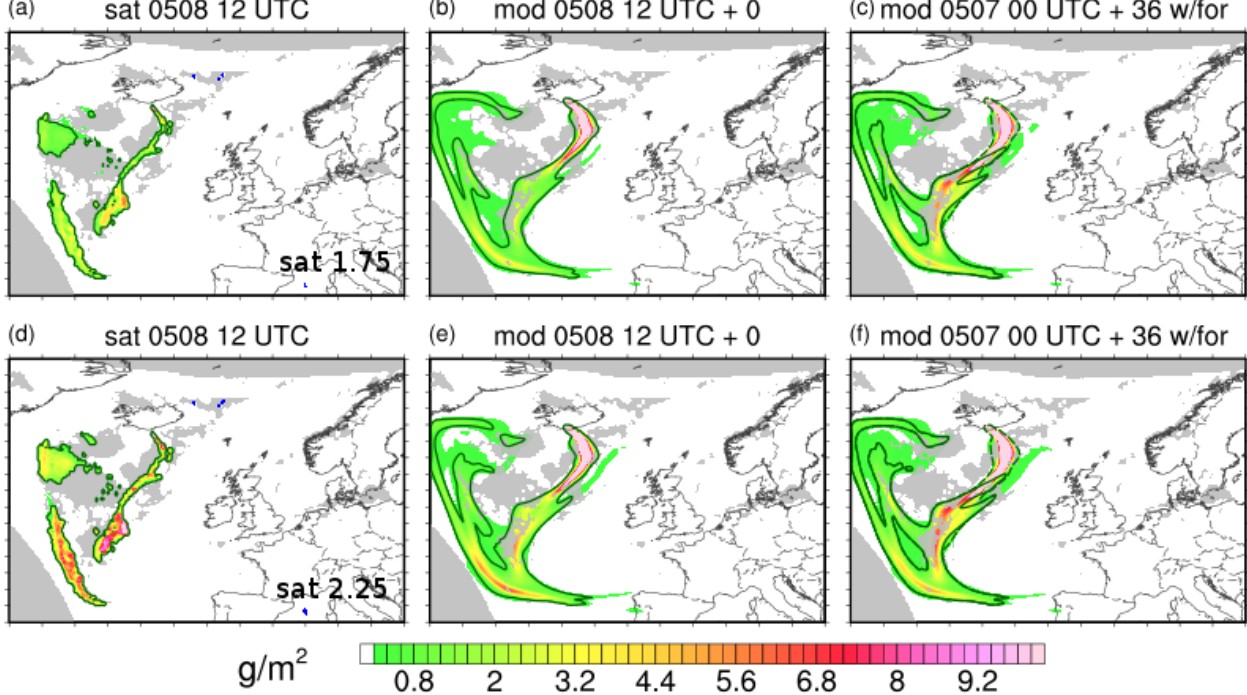

**Figure 10: The same as Figure 9 but for 8 May 12 UTC, showing the satellite data (left) and a posteriori model simulations for the first forecast hour (middle) and 36 hour forecasts (right) with inversions for satellite retrieval data with the 1.75 (top row) and 2.25 (bottom row) size distribution assumption.**







**Figure 11: SAL (Structure Amplitude Location) results for model simulations run with the a priori emissions calculated for the 1.75 (left) and 2.25 (right) satellite retrieval (top row), and results for the model simulations using a posteriori emissions started every 12 hours from 6 May 00 UTC to 9 May 00 UTC with a 48 hour forecast using either a forecast emission estimate (dashed lines) or a zero ash emission term in the forecast (straight lines). Grey areas show the assimilation period where the emission estimate is calculated by the inversion. Model simulations with the 1.75 a posteriori source term are compared to the 1.75 satellite observation field, and those with the 2.25 a posteriori emissions are compared to 2.25 satellite retrievals.**