# Peer review of "Uncertainty assessment and applicability of an inversion method for volcanic ash forecasting"

_Atmospheric Chemistry and Physics, 2016_

## Referee Comment (RC1) · Anonymous Referee #2 · 2 Mar 2017

General comments:

This paper concerns the forecasting of volcanic ash plumes, and presents a case study of the 2010 Eyjafjallajökull eruption. The main issue dealt with in the paper is the optimal estimation of the "source term", that is, an estimate of the injection of ash as a function of height and time, based on a combination of a priori information, an ash transport model (fed by meteorological reanalysis data) and satellite retrievals of the ash plume. While other papers have pioneered the data assimilation technique used to estimate the ash emission, this work focuses on a forecasting scenario, where the assimilation is used as an initialization for an ash transport forecast simulation. The inversion technique which is used to estimate the volcanic emissions relies not just on

the a priori information and satellite retrievals, but also on estimates of the uncertainty in these values: in general, any optimal estimation or assimilation technique weights different sources of information based on their relative uncertainties. The major focus of the study is on the impact of different uncertainties on the source term estimation, and on the results of the forecasts.

The subject of this work is appropriate for ACP. In general, the organization, structure, and writing could be improved to help the reader understand the major results and conclusions of the work. References to prior works appear to be appropriate, and the methods are described enough that it is possible to follow the logic of the numerical experiments. The conclusions are generally supported by the results (some comments below), and the study appears to be a tangible step towards robust volcanic ash evolution prediction systems.

One of the major results of the paper appears to be that uncertainty in the width of the log-normal ash size distribution assumed in the satellite retrievals is the major uncertainty in the source term estimation technique–at least that this uncertainty is much more important than so called "other-than-size" uncertainties in the retrieval. This is presented as a rather general conclusion. I see two problems with this conclusion.

Firstly, and generally, it is hard to make concrete conclusions about the impact of different uncertainties when the ranges of uncertainties used in the study for the different parameters are rather arbitrarily, and not uniformly sampled. Uncertainties in "other-than-size" parameters in the retrievals are sampled at 0-200%, while the uncertainty in the a priori is sampled from 25-100%. Ash size distribution widths are sampled at a set of discrete values, and it's not clear how any of these ranges compare to the fundamental uncertainties in these physical parameters. It is therefore hard, or impossible, to draw conclusions about the relative importance of different uncertainties in the results of the inversion.

Secondly, it is hard to believe that changing the uncertainties in the retrieved ash mass

(the "other-than-size" satellite parameters) from 0 to 200% (Pg 7, line 1-2) will have such small impact on the estimated source term (as in table 1). The authors note that part of the explanation for this may be because the a priori used in the emission estimation is rather large compared to all the a posteriori results. So, if the uncertainty used for the a priori is too small, then the assimilation might be too strongly constrained by the a priori, and the satellite observations have a similar impact on the result no matter the uncertainty assigned to them. If this is the case, then it is a result which is very specific to this case–the opposite result might occur if the a priori is very realistic. This point needs to be very carefully considered in the abstract, results, and conclusions. It would actually be extremely useful to repeat the analysis using the 0.1 fine ash fraction, which would apparently greatly improve the accuracy of the a priori.

The text is often hard to understand, partly because many different names are used interchangeably for the same things. For example, the ash emissions as a function of height and time which are estimated by the "inversion technique" are referred to as the "source term", the "source emission term", the "source estimate", the "emission estimate" and so on. If the same thing is referred to in each case, then the same name should be used.

Specific comments

Pg 1, l7: Data assimilation techniques are not a development in and of themselves, but the application or use of them in this field may be.

Pg 1, l7: The aim or advantage of data assimilation is a little more subtle than just bringing model results in close agreement with observations. When used properly, data assimilation results provide more value than the model results or observations individually.

Pg 1, l19: Varying assumptions in the satellite retrieval only translates to uncertainties in the estimated emissions if those variations correspond to actual uncertainties in the satellite retrieval, ideally quantified in some systematic way.
Pg 1, l20: It's not clear what "weighting of uncertainties" means here. Uncertainties are used by data assimilation to weight the relative contributions of different sources of information, but the uncertainties are not themselves weighted.

Pg 2, l6: It's not clear how the use of the a posteriori emissions reduces uncertainties connected to the satellite observations.

Pg 2, l7: isn't the forecast of real interest that of the ash transport? Forecasting the ash emission seems like a very different, and difficult problem.

Pg 2, l14: "source term" should be well defined in its first use.

Pg 2, l26: It wasn't clear to me at first that the "zero" and "constant" a priori estimates were two different things. Also, this result likely depends on the uncertainty assigned to the a priori, and the size of the assumed "constant" a priori.

Pg 3, l1: this sentence seems to contradict the previous sentence.

Pg 3, l15: It should be made clear that this list is not exhaustive, certainly one could use other estimates in the forecast, including an upper limit.

Pg 3, l17: It's not really clear to me why or how using the average of the past few hours in the forecast "limits the uncertainty" of the emissions: this implicitly assumes that the volcanic emissions are most likely to persist at a relatively constant rate. If this can really be shown to be the best assumption, then it would be an interesting result.

Pg 4, l15: Is there a justification for this threshold?

Pg 5, l32: Are the satellite measurements hourly means of many satellite measurements, or single "snapshots"?

Pg 7, l1-2: If it is true that the so called "other-than-size" uncertainty is simply an uncertainty associated with the retrieved ash mass, it would be clearer to refer to it as so ,i.e., "mass loading uncertainty" or so. I see that the assumed size distribution also impacts the mass loading, and so there is an argument that the term "mass loading

uncertainty" may not be exactly unique, but if it more clearly describes what it actually is, then I think the process will be easier for the reader to understand.

Pg 7, l27: This is confusing, if the satellite observations are used as the observation field, won't this analysis compare the model forecast with the satellite observations? And if so, isn't the answer obvious, that is when the uncertainty assumed for the satellite observations is relatively small compared to the other uncertainties, then the assimilation will put more weight on the satellite obs and produce a closer agreement?

Pg 9, l18: In Figure 5b, it's not clear which color refers to which a priori uncertainty value.

Pg 9, l19: Does this last sentence of the paragraph refer to Fig 5a or 5b? And does the spread produced by using different size assumptions in the satellite retrieval represent the full a priori uncertainty, or could other sources of uncertainty also affect it?

pg 10, l7: I don't think that using a range of uncertainty values in the operation forecasting is ideal: in fact, the ideal forecasting should use the most accurate estimates of the real uncertainties as possible: the result of the forecast should then produce the most accurate result.

Pg 10, l13: What does "typical" mean in this case, is it similar to the overall ensemble mean?

Pg 12, l14: The term "optimized field" is not easy to understand, is this simply the forecast model result?

Pg 16, l15: This wasn't really an operational forecast setting, more a kind of hindcast scenario.

Pg 17, l21: I don't think the paper really shows that the use of the emission inversion technique produces improved confidence in the satellite data.

Editorial comments

Pg 1, l11: exploits->explores

Pg 1, l13: it may couple measurements from a satellite instrument, but not the satellite itself.

Pg 2, l10: should probably be clear you refer to airplane windshields.

Pg 2, l12: current->instantaneous

Pg 2, l21: suggest: "weight their relative contributions to the inversion results"

Pg 2, l32: A more accurate emission term is. . .

Pg 4, l15: "Emitted" from an emission? Perhaps "resulting from a unit ash emission" is closer to the mark?

Pg 5, l32: I'm not sure what "forward mean" means.

Pg 7, l13: 0.4 is

Pg 9, l28: "trough"?

Pg 9, l 32: "left-most"

Pg 11, l23: the change in a posteriori emissions. . . is similar. . .

Pg 11, l24: April and May periods

Pg 16, l15: The start of the conclusions shouldn't reference "the inversion method"—a reader might not have necessarily read the preceding sections in detail.

Pg 16, l20: The observed ash cloud. . . is shown to be difficult. . .

Pg 16, l25: The retrieval does not really "decide", maybe "distinguish" is a better word choice.

Pg 17, l11: I don't think the "times" are reduced.

Pg 17, l17: . . .the model. . . has more ash (although, the model doesn't really "have"

anything).

---

## Short Comment (SC1) · 30 Mar 2017

General comments:

The paper treats an important issue which has become topical for the entire meteo/volcanological community since the eruption at Eyjafjallajökull (Iceland) in 2010. The argument is treated by using proper approach and state-of-the-art numerical modelling. Overall the paper merits attention, but there is a key element that should be better treated and discussed before building the conclusions.

It is well known the problem of constraining and quantifying the eruptive source conditions when we need to model and, eventually, forecasts the dispersal of volcanic ash

cloud in the atmosphere. It is so important that an approach to obtain them is by inverting the semi-quantitative information retrieved by satellite images. On the other hand in this paper the a-priori scenario is built on data obtained by applying the Mastin et al. (2009) relationship to the plume height radar observations in order to assess a-priori values of the mass flow rate over the entire eruption. But several published papers clearly stated that the first few days of the eruption were characterized by a very strong wind that bent down the plume (see Petersen at al. 2012). In such conditions the Mastin et al. formula cannot be applicable because would underestimate the mass flow rate. Actually, revised mass flow rates obtained from both empirical and modelling approaches have been published in Gudmundsson et al. 2012 (for the entire eruption) and in Folch et al. 2011 (for the first week of activity) and both agree on mass flow rate values for the first days of the eruption to be on the scale of 10E6 kg/s. This paper, by assuming Mastin formula, get values for the mass flow rate up to 3x10E5 kg/s (Fig. 3, section 2.4). This would imply one third of the erupted mass has been assumed in input for the model simulations performed for this paper. So, my question here is: 1) why not to use the best assessment for mass flow rate when it is available and already published?, 2) how the results and the conclusion of this paper would change if the a-priori scenario would be built on these "more constrained" values of mass flow rates? I think the authors should comment to these questions to make the methodology and the conclusion more robust.

Additionally, Mastin et al. 2009 paper is adopted to assess the fraction of fine ash and as explained in section 2.2 the simulations in this paper have been using ash particle bins from 4 to 25 micron. So, the question here is: how did you extrapolate the information provided by Mastin et al. 2009 which provide an indication for fraction of material smaller than 63micron to assess the amount of ashes smaller than this size?

---

## Referee Comment (RC2) · R. Denlinger (Referee) · 1 Apr 2017

I have reviewed this paper for a second time, and found that my original concerns were addressed. The paper reads well and makes a contribution to forecasting that should be published.

---

## Author Comment (AC1) · 26 May 2017

Response to Review #2

We thank the reviewer for taking the time to thoroughly understand the work presented in this study, and for the helpful comments and suggestions for improving the manuscript.

Answers to the general comments are given below, followed by point by point answers to the Specific and Editorial comments. Reviewer comments are given in black, answers are given in blue, and changes in the manuscript are noted in quotations (""), also in blue.

General comments:

One of the major results of the paper appears to be that uncertainty in the width of the log-normal ash size distribution assumed in the satellite retrievals is the major uncertainty in the source term estimation technique–at least that this uncertainty is much more important than so called "other-than-size" uncertainties in the retrieval. This is presented as a rather general conclusion. I see two problems with this conclusion.

Firstly, and generally, it is hard to make concrete conclusions about the impact of different uncertainties when the ranges of uncertainties used in the study for the different parameters are rather arbitrarily, and not uniformly sampled. Uncertainties in "other-than-size" parameters in the retrievals are sampled at 0-200%, while the uncertainty in the a priori is sampled from 25-100%. Ash size distribution widths are sampled at a set of discrete values, and it's not clear how any of these ranges compare to the fundamental uncertainties in these physical parameters. It is therefore hard, or impossible, to draw conclusions about the relative importance of different uncertainties in the results of the inversion.

The selection of uncertainty estimates are based upon what have been used in previous inversion studies cited in the manuscript. Although the Kelut study (Kristiansen et al., 2015) applies an a priori uncertainty of 1000 %, the other studies cited in the manuscript use uncertainty estimates for the a priori source from 100 % and downwards. Mass load uncertainty estimates for the satellite data are difficult to set and are often not well documented in the literature. Wen and Rose (1994) and Corradini et al. (2008) estimate errors from 40 - 60 %, however as seen in the satellite data used in this study as well as in the other studies cited, the uncertainty estimates can have a more extensive range. Therefore a wider range of 0 – 200% is used for mass load uncertainty in this study.

The geometric standard deviation of the particle size distribution is chosen as a subset of what is presented in Francis et al. (2012). There are relatively few observations of size distribution of ash particles, and arguably these can also be only representative for the observed time and area as plume dynamics and transport conditions have impact on fine ash aggregations and hence the size distribution. Schumann et al. (2011) and Johnson et al. (2012) presents some in situ measurements of particle size distribution made during the Eyjafjallajökull 2010 eruption. Johnson et al. (2012) show that the size distribution of ash particles measured with Cloud and Aerosol Spectrometer (CAS) (0.6 to 35 μm) for six flights between 4 to 18 May 2010 may be described by lognormal distributions with standard deviations from 1.8 to 1.9. Gasteiger et al. (2011) use a standard deviation range from 1.2 to 4 supporting the use of larger and smaller standard deviations (1.5, 2.25) as well.

Even though the different uncertainties are not uniformly sampled it does show the spread in the results based on realistically set uncertainties. However more studies ideally should be done for different volcanic eruptions to enable a more complete conclusion on the uncertainty dependency for the a posteriori, to be drawn.

Secondly, it is hard to believe that changing the uncertainties in the retrieved ash mass (the "other-than-size" satellite parameters) from 0 to 200% (Pg 7, line 1-2) will have such small impact on the estimated source term (as in table 1). The authors note that part of the explanation for this may be because the a priori used in the emission estimation is rather large compared to all the a posteriori results. So, if the uncertainty used for the a priori is too small, then the

assimilation might be too strongly constrained by the a priori, and the satellite observations have a similar impact on the result no matter the uncertainty assigned to them. If this is the case, then it is a result which is very specific to this case– the opposite result might occur if the a priori is very realistic. This point needs to be very carefully considered in the abstract, results, and conclusions. It would actually be extremely useful to repeat the analysis using the 0.1 fine ash fraction, which would apparently greatly improve the accuracy of the a priori.

The authors agree that the result depend on the reduction of the a priori estimate. We repeated the analysis using the 0.1 fine ash fraction and the results are shown in Figure 1.

[Figure]

**Figure 1: Spread of the a posteriori for 0.1 fine ash fraction for the four satellite data sets with the four different size distribution assumptions (sat 1.5 – 2.25). (Left) The spread in the a posteriori caused by varying the uncertainty connected to the satellite data, with a priori uncertainty set to 75 %. (Right) The spread in a posteriori caused by varying the a priori uncertainty, with a constant satellite uncertainty of 100 %.**

The results are similar to those presented in the manuscript, although due to the 0.1 fine ash fraction the spread in the a posteriori solutions for the sensitivity to a priori uncertainty exhibit some differences. Compared to the a posteriori for the 0.4 fine ash fraction, the spread for sat 1.5 and 1.75 is smaller at 5 km height as the satellite retrievals are more similar to the a priori at these heights. There is also a larger spread for the sat 2 and sat 2.25 retrieval compared to the 0.4 fine ash retrieval.

For the mass loading uncertainty, since the uncertainty of the satellite retrieval is a percent of the retrieved ash mass loading, the spread in a posteriori estimates for the satellite retrieval with geometric standard deviation of 1.5 is much smaller than the one with a geometric standard deviation of 2.25. However, similar to the plot in the manuscript, the a posteriori spread due to the mass load satellite uncertainty is still smaller than the a posteriori spread caused by using the different satellite data set. indicating that although using a large spread for mass load uncertainty (up to 200%) it does not reflect the real uncertainty in the satellite retrieval as seen in the larger spread between the sat 1.5 and sat 2.25 a posteriori results.

Even with using the 0.1 fine ash fraction, the solutions show that the a priori uncertainty contributes the most to the sensitivity of the a posteriori even with a smaller spread (25-100% vs. 0-200%). To include this in the manuscript, the sentence on line 24 page 1 will be changed to:

"Setting large uncertainties connected to both the a priori and the satellite uncertainties are shown to compensate each other, but the a priori uncertainty is found to be most sensitive."

Line 29 page 16 will be changed to:

"The spread in a posteriori due to the a priori uncertainty for the four satellite retrievals is largest where the a posteriori and a priori deviate the most. Mass loading uncertainties connected to the satellite retrieval are found to have smaller effect."

The text is often hard to understand, partly because many different names are used interchangeably for the same things. For example, the ash emissions as a function of height and time which are estimated by the "inversion technique" are referred to as the "source term", the "source emission term", the "source estimate", the "emission estimate" and so on. If the same thing is referred to in each case, then the same name should be used.

The authors agree that using the same name should be used throughout the text. In the revised manuscript we consistently use "source term".

Specific comments

Pg 1, l7: Data assimilation techniques are not a development in and of themselves, but the application or use of them in this field may be.

Answered below, pg 1, l7.

Pg 1, l7: The aim or advantage of data assimilation is a little more subtle than just bringing model results in close agreement with observations. When used properly, data assimilation results provide more value than the model results or observations individually.

The authors agree and the sentence has been changed to:

"One major development has been the application of data assimilation techniques, which combine models and satellite observations such that an optimal understanding of ash clouds can be gained."

Pg 1, l19: Varying assumptions in the satellite retrieval only translates to uncertainties in the estimated emissions if those variations correspond to actual uncertainties in the satellite retrieval, ideally quantified in some systematic way.

The study finds that the assumptions made in the satellite retrieval is more important than the associated mass load uncertainty as the inversion constrains the source term by the satellite observations. The real size distribution in the satellite observation is unknown at the start of a forecast, as well as the pixels may be contaminated by the presence of water and ice clouds. To quantify the uncertainty in the satellite retrievals is beyond the scope of this study and have been done in several studies mentioned above.

The sentence is changed to:

"Varying the assumptions made in the satellite retrieval is seen to affect the a posteriori emissions and modelled ash column loads, and modelled column loads therefore have uncertainties connected to them depending on the uncertainty in the satellite retrieval."

Pg 1, l20: It's not clear what "weighting of uncertainties" means here. Uncertainties are used by data assimilation to weight the relative contributions of different sources of information, but the uncertainties are not themselves weighted.

The authors agree that this is not the correct formulation. The sentence has been changed as follows.

"By further exploring our uncertainty estimates connected to a priori emissions and the mass load uncertainties in the satellite data, the uncertainty in the a priori estimate is found in this case to have an order of magnitude more impact on the a posteriori solution compared to the mass load uncertainties in the satellite data."

Pg 2, l6: It's not clear how the use of the a posteriori emissions reduces uncertainties connected to the satellite observations.

Please see answer below, p.2, l7.

Pg 2, l7: isn't the forecast of real interest that of the ash transport? Forecasting the ash emission seems like a very different, and difficult problem.

Thank you for noticing this. We have clarified this sentence as follows:

"Overall, using the a posteriori emissions in our model reduces the uncertainties in the ash plume forecast, because it corrects effectively for false positive satellite retrievals, temporary gaps in observations, and false a priori emission estimates in the window of observation."

Pg 2, l14: "source term" should be well defined in its first use.

Agreed, we have added a better definition, and use 'source term' throughout the manuscript.

Sentence replaced by:

"These models need a robust estimate of source parameters such as ash release height, amount of ash released and ash particle sizes. The combined choice of these source parameters is below named the source term. During an eruption, information about the source term is often limited."

Pg 2, l26: It wasn't clear to me at first that the "zero" and "constant" a priori estimates were two different things. Also, this result likely depends on the uncertainty assigned to the a priori, and the size of the assumed "constant" a priori.

Thank you for pointing out that this is not clear, and the authors hope that a rewrite will help:

"Eckhardt et al. (2008) found small differences between the a posteriori estimates when using a zero value or a non-zero constant value as a priori estimate for the Jebel at Tair 2007 eruption."

Pg 3, l1: this sentence seems to contradict the previous sentence.

The authors agree that the sentence was a contradiction, and the last part of the sentence was not needed as applying more satellite observations reduces the forecast time and represents the same results as younger plumes. The sentence is changed to:

"A better source term for the entire episode studied may be found by assimilating several satellite observations over the entire period studied."

Pg 3, l15: It should be made clear that this list is not exhaustive, certainly one could use other estimates in the forecast, including an upper limit.

The authors agree, and the sentence is changed to:

"For emissions to be used during the forecast period there are several possibilities for example 1) assume no further emissions; 2) use the latest a priori emission from Mastin et al. (2009); or 3) use the average of the last hours of the a posteriori from the inversion."

Pg 3, l17: It's not really clear to me why or how using the average of the past few hours in the forecast "limits the uncertainty" of the emissions: this implicitly assumes that the volcanic emissions are most likely to persist at a relatively constant rate. If this can really be shown to be the best assumption, then it would be an interesting result.

The emission in the forecasting period is difficult and using the average of the last hours will include some scaling from the satellite data. The study shows that the results may improve if the latest emission estimate is included in the forecast, it is also beneficial for the forecast to include it in the case where the emissions have ceased. The sentence is changed to:

"Assuming the eruption continues, the latter option includes some information from the satellite observations that may limit the uncertainty of using a priori default emissions. The use of an average of the emission during the last12 hour will be compared  here against  zero emissions in the forecast period."

Pg 4, l15: Is there a justification for this threshold?

The threshold is chosen pragmatically to not include too much model data with low values, which will increase the computational time as well as the limitation of points that the inversion technique can handle with the current set up. This is a problem for Eulerian models and not for Lagrangian model such as FLEXPART, for which the inversion method was originally designed for. The limit we have chosen reduced in our case the data volume by around 30%. Visual inspection of the volcanic plumes assured us of capturing also well the borders of the plume.

Pg 5, l32: Are the satellite measurements hourly means of many satellite measurements, or single "snapshots"?

The satellite data are snapshots.

Pg 7, l1-2: If it is true that the so called "other-than-size" uncertainty is simply an uncertainty associated with the retrieved ash mass, it would be clearer to refer to it as so ,i.e., "mass loading uncertainty" or so. I see that the assumed size distribution also impacts the mass loading, and so there is an argument that the term "mass loading uncertainty" may not be exactly unique, but if it more clearly describes what it actually is, then I think the process will be easier for the reader to understand.

The term for this uncertainty was difficult to decide as there are many uncertainties for the satellite data, firstly related to different retrieval details in particular cloud and surface corrections, and secondly those reflecting the uncertainties in the retrieved mass loading values.

"To see the effect of the mass loading uncertainties on the inversion calculations, four uncertainties are assigned to the satellite data in separate inversion calculations; 0 %, 50 %, 100 % and 200% as a percent of the retrieved column load in each grid cell."

Pg 7, l27: This is confusing, if the satellite observations are used as the observation field, won't this analysis compare the model forecast with the satellite observations? And if so, isn't the answer obvious, that is when the uncertainty assumed for the satellite observations is relatively small compared to the other uncertainties, then the assimilation will put more weight on the satellite obs and produce a closer agreement?

Yes, its true, the satellite observations used here for evaluation are not independent of the assimilation. The reason for

using the satellite observations again, is that they are the only data containing spatio-temporal info. The evaluation also shows that the assimilated fields are not perfect and the quality depends on the amount of satellite data assimilated. For the forecast period we can further assume that the satellite data represent an independent estimate of the ash cloud.   We are not proving that assimilation and satellite data are more similar for different mass load uncertainties; we test instead, how good the agreement between assimilated field and satellite is for different model simulations using more and more data, or different satellite data. For this part of the manuscript, the reference uncertainties are set for a priori and mass load uncertainties on the 1.75 and 2.25 sat data. The text has been clarified.

"The satellite data used for the inversion are also used for calculating the SAL scores. One advantage is the broad spatial coverage of the satellite data. While this does not allow a totally independent check of the assimilation, it provides information on how much the different amounts of satellite data entering the inversion procedure influence the performance in the observed period and in the forecast period. In particular in the forecast period, the satellite data are rather independent from the inversion assimilation."

Pg 9, l18: In Figure 5b, it's not clear which color refers to which a priori uncertainty value.

The colors refer to the different satellite data sets used as shown in the legend. The sentence describing the figure has been clarified:

"In figure 5b) the mass loading satellite data uncertainty is set to 100% and the a priori emission uncertainty is varied from 25 to 100% for each of the four satellite sets."

Pg 9, l19: Does this last sentence of the paragraph refer to Fig 5a or 5b? And does the spread produced by using different size assumptions in the satellite retrieval represent the full a priori uncertainty, or could other sources of uncertainty also affect it?

The sentence refers to fig 5b. Indeed, as not all uncertainties for the a priori are studied here the sentence is changed to:

"The resulting spread in vertical emission distribution for the different satellite data sets represents the a priori uncertainties studied here."

pg 10, l7: I don't think that using a range of uncertainty values in the operation forecasting is ideal: in fact, the ideal forecasting should use the most accurate estimates of the real uncertainties as possible: the result of the forecast should then produce the most accurate result.

Using a range is not ideal and indeed not feasible, as we state in the paragraph. What we try to investigate is, how accurate the uncertainty estimate must be, and how accurate combined uncertainties must be. The uncertainty of the uncertainties is influencing the end result. So what to chose, how exact do we have to estimate the uncertainty? This is one of the questions the study is addressing. What to do at the start of an eruption when little information about the uncertainties is known? The results presented here may provide some insight how exact we have to estimate the uncertainties used in the inversion method until further information is known. The second part of the paragraph is changed to:

"Using a range of uncertainty values is probably not feasible in an operational setting. But the results presented  here provide insight into the impact of the uncertainties on the resulting spread of the a posteriori source term. It may guide operational efforts in the case of future volcanic eruptions to establish a combination of realistic uncertainty estimates, not being unnecessarily over precise on individual uncertainties."

Pg 10, l13: What does "typical" mean in this case, is it similar to the overall ensemble mean?

Typical means here the estimates closest to what is used in previous studies and what is found to give reasonable result within the spread of a posteriori source terms. The word typical might not be the best word though, and the sentence is changed to:

"The inversion result and associated simulation is our best guess and a reference for comparing our different experiments."

Pg 12, l14: The term "optimized field" is not easy to understand, is this simply the forecast model result?

Optimized field refers to the model ash field with an a posteriori source term that includes all the satellite observations up until the start time of the forecast. This is previously referenced as initially simulated ash concentrations and should be referenced this way again:

"The model simulation does not manage to transport narrow ash clouds with high concentrations due to numerical diffusion and the initially simulated concentrations (Fig 9b) therefore have smaller maximum values. "

Pg 16, l15: This wasn't really an operational forecast setting, more a kind of hindcast scenario.

The authors agree that forecast setting is not correct wording as it is a hindcast, however the test is done in a setup that resembles what could be done if a volcanic eruption occurred in the future. The sentence is changed to:

"In this paper an inversion method for source term calculations is tested in an operational forecasting setup over two short periods of four days during the Eyjafjallajökull 2010 eruption, simulating a forecast situation."

Pg 17, l21: I don't think the paper really shows that the use of the emission inversion technique produces improved confidence in the satellite data.

Figure 9 and 10 shows areas where the satellite retrieval find ash clouds that could not be simulated with the model, These areas are difficult to label false positives without also studying the transport of emission from the erupting volcano. This is done by the dispersion model with ash released at several heights. The authors agree that confidence is not the right word, and the sentence is changed accordingly:

"Model results with a posteriori emissions decrease the ambiguity when using both the forecast and the satellite observations by obtaining model ash loads more comparable to satellite values, and facilitating the interpretation of the satellite data by identifying areas with e.g. false positives or undetected ash."

Editorial comments

Pg 1, l11: exploits->explores

Changed accordingly.

Pg 1, l13: it may couple measurements from a satellite instrument, but not the satellite itself.

Thank you for noticing this, the sentence is changed to:

".., which is computed by an inversion method that couples the satellite retrievals and a priori emissions with dispersion model data."

Pg 2, l10: should probably be clear you refer to airplane windshields.

Changed accordingly.

Pg 2, l12: current->instantaneous

Changed accordingly.

Pg 2, l21: suggest: "weight their relative contributions to the inversion results"

Changed accordingly.

Pg 2, l32: A more accurate emission term is. . .

Changed accordingly.

Pg 4, l15: "Emitted" from an emission? Perhaps "resulting from a unit ash emission" is closer to the mark?

Changed accordingly.

Pg 5, l32: I'm not sure what "forward mean" means.

This is a forward interpolation method that takes the average over the matching input cells. The sentence is changed to:

"For the inversion, satellite observations for every hour are used as input and forward interpolated to the 0.25 x 0.25 degree model domain and if two or more pixels belong to the same grid cell the column loads are averaged."

Pg 7, l13: 0.4 is

Changed accordingly.

Pg 9, l28: "trough"?

Changed accordingly.

Pg 9, l 32: "left-most"

Changed accordingly.

Pg 11, l23: the change in a posteriori emissions. . . is similar. . .

Changed accordingly.

Pg 11, l24: April and May periods

Changed accordingly.

Pg 16, l15: The start of the conclusions shouldn't reference "the inversion method"ãA˘Ta˘ reader might not have necessarily read the preceding sections in detail.

Changed to:

"In this paper an inversion method for source term calculations is tested.."

Pg 16, l20: The observed ash cloud. . . is shown to be difficult. . .

Changed accordingly.

Pg 16, l25: The retrieval does not really "decide", maybe "distinguish" is a better word choice.

Changed accordingly.

Pg 17, l11: I don't think the "times" are reduced.

Changed to:

"Emission at times with no significant ash emissions is…"

Pg 17, l17: . . .the model. . . has more ash (although, the model doesn't really "have" anything).

Changed to:

"The SAL scores show that model results at most times have more."

References:

Corradini, S., Spinette, C., Carboni, E., Tirelli, C., Buongiorno, M. F., Pugnaghi, S., and Gangale, G., Mt. Etna tropospheric ash retrieval and sensitivity analysis using Moderate Resolution Imaging Spectroradiometer Measurements, J. of Applied Remote Sensing, 2, 1, 023550-023550-20, doi:10.1117/1.3046674, 2008.

Francis, P. N., Cooke, M. C., and Saunders, R.W.: Retrieval of physical properties of volcanic ash using Meteosat: A case study from the 2010 Eyjafjallajökull eruption, Journal of Geophysical Research: Atmospheres, 117, doi:10.1029/2011JD016788, URL http://dx.doi.org/10.1029/2011JD016788, 2012.

Gasteiger, J., Groß, S., Freudenthaler, V., & Wiegner, M: Volcanic ash from Iceland over Munich: mass concentration retrieved from ground-based remote sensing measurements. *Atmospheric chemistry and physics*, *11*(5), 2209-2223, 2011.

Johnson, B., Turnbull, K., Brown, P., Burgess, R., Dorsey, J., Baran, A. J., ... & Hesse, E: In situ observations of volcanic ash clouds from the FAAM aircraft during the eruption of Eyjafjallajökull in 2010. *Journal of Geophysical Research: Atmospheres*, *117*(D20), 2012.

Kristiansen, N. I., A. J. Prata, A. Stohl, and S. A. Carn, Stratospheric volcanic ash emissions from the 13 February 2014 Kelut eruption, Geophys. Res. Lett., 42, 588–596, doi:10.1002/2014GL062307, 2015.

Prata, A. J., and A. T. Prata (2012), Eyjafjallajökull volcanic ash concentrations determined using Spin Enhanced Visible and Infrared Imager measurements, J. Geophys. Res., 117, D00U23, doi:10.1029/2011JD016800.

Schumann, U., Weinzierl, B., Reitebuch, O., Schlager, H., Minikin, A., Forster, C., Baumann, R., Sailer, T., Graf, K., Mannstein, H., Voigt, C., Rahm, S., Simmet, R., Scheibe, M., Lichtenstern, M., Stock, P., Rüba, H., Schäuble, D., Tafferner, A., Rautenhaus, M., Gerz, T., Ziereis, H., Krautstrunk, M., Mallaun, C., Gayet, J.-F., Lieke, K., Kandler, K., Ebert, M., Weinbruch, S., Stohl, A., Gasteiger, J., Groß, S., Freudenthaler, V., Wiegner, M., Ansmann, A., Tesche, M.,

Olafsson, H., and Sturm, K.: Airborne observations of the Eyjafjalla volcano ash cloud over Europe during air space closure in April and May 2010, Atmos. Chem. Phys., 11, 2245-2279, doi:10.5194/acp-11-2245-2011, 2011.

Wen, S. and Rose, W. I.: Retrieval of sizes and total masses of particles in volcanic clouds using AVHRR bands 4 and 5, J. Geophys. Res., 99, 5421–5431, 1994.

---

## Author Comment (AC2) · 26 May 2017

Response to Short Comment #1

We thank Sara Barsotti for taking the time and posting this short comment which we think would improve our manuscript.

Answers to the questions are given below, questions are given in black, answers are given in blue, and changes in the manuscript are noted in quotations (""), also in blue.

1) why not to use the best assessment for mass flow rate when it is available and already published?,

If a new volcanic eruption erupted now, there would be little information on the flow rate available during the first days, and the Mastin et al. (2009) relationship is the first best guess used in a . As this study aims to represent a real case, the Mastin relationship is used, and the improvement of this simple relationship should be done with the inversion technique.

2) how the results and the conclusion of this paper would change if the apriori scenario would be built on these "more constrained" values of mass flow rates?

Figure 1 in reply to review #2 shows inversion results for the four day period using the 0.1 fine ash fraction. The sensitivity of the a posteriori with regards to uncertainties connected to a priori and satellite mass load re shown to be similar only the a priori now have similar values as the a posteriori with the smaller standard deviation 1.5, 1.75. The sensitivity spread between the a posteriori with different uncertainty still do not represent the real uncertainty in the satellite retrieval as seen in the different a posteriori obtained by using different satellite sets.

Comparing to the estimates from Folch et al. (2011) and Gudmundsson at al. (2012) where larger source terms were found indicates that there indeed is ash that is not observed by the satellite and, especially during the April period where the satellite retrieve only narrow clouds with high ash loads. The corresponding model fields show clouds more spread out with lower concentrations.

This aspect should be included in the discussion of the manuscript:

p.15 line 26:

"Even though the 0.1 fine ash fraction match better with satellite retrievals, Gudmundsson et al. (2012) found by studying ash deposition on land almost four times more very fine ash ($< 28$ μm) for the first days of the Eyjafjallajökull eruption (14-16 April) compared to Stohl et al. (2011) a posteriori over the entire eruption. This large discrepancy indicates that satellite observations indeed do not observe all ash that is either obscured by meteorological clouds or too opaque ash clouds."

how did you extrapolate the information provided by Mastin et al. 2009 which provide an indication for fraction of material smaller than 63micron to assess the amount of ashes smaller than this size?

The Mastin fine ash fraction is distributed only over 4 to 25 μm as these are the sizes that the satellite is most sensitive to and are not extrapolated. This may miss cause some of the smaller and larger ash articles not to be described correctly in the model transport. Other size distributions, such as the one used by London VAAC (Volcanic Ash Advisory Centre) described in Hobbs et al. (1991) show that 95.6 % of the measured ash distribution is under 30 μm. However, more work should be done on how to translate the sizes that is sensitive to satellite data to the larger (μm $> 30$) and smaller (μm $< 4$) sizes.

References:

Folch, A., Costa, A., & Basart, S: Validation of the FALL3D ash dispersion model using observations of the 2010 Eyjafjallajökull volcanic ash clouds. *Atmospheric Environment*, *48*, 165-183, 2012.

Gudmundsson, M. T., Thordarson, T., Höskuldsson, Á., Larsen, G., Björnsson, H., Prata, F. J., ... & Hayward, C. L. (2012). Ash generation and distribution from the April-May 2010 eruption of Eyjafjallajökull, Iceland. *Scientific reports*, *2*, 572, 2012.

Hobbs, P. V., Radke, L. F., Lyons, J. H., Ferek, R. J., Coffman, D. J., & Casadevall, T. J: Airborne measurements of particle and gas emissions from the 1990 volcanic eruptions of Mount Redoubt. *Journal of Geophysical Research: Atmospheres*, *96*(D10), 18735-18752, 1991.

Mastin, L. G., Guffanti, M., Servranckx, R., Webley, P., Barsotti, S., Dean, K., ... & Schneider, D: A multidisciplinary effort to assign realistic source parameters to models of volcanic ash-cloud transport and dispersion during eruptions. *Journal of Volcanology and Geothermal Research*, *186*(1), 10-21, 2009.

---

## Author Comment (AC3) · 26 May 2017

We thank the reviewer for taking the time to read our manuscript and for the original review which improved our manuscript.